# Theoretical Investigation of Adafactor for Non-Convex Smooth Optimization

**Yusu Hong**
Center for Data Science
and School of Mathematical Sciences
Zhejiang University
yusuhong@zju.edu.cn

**Junhong Lin**[*]
Center for Data Science
Zhejiang University
junhong@zju.edu.cn

## Abstract

Adafactor is an early memory-efficient optimization algorithm proposed as an alternative to Adam. By eliminating first-order momentum and employing a rank-1 matrix factorization to approximate the second-moment matrix, Adafactor achieves near-zero memory overhead compared to traditional gradient descent methods. Despite its practical suitability for large-scale training tasks where memory efficiency is critical, its theoretical convergence analysis remains unexplored, largely due to the challenges posed by its matrix factorization and update clipping mechanisms. In this work, we provide a convergence analysis of Adafactor for non-convex smooth optimization. We establish optimal convergence rates (up to logarithmic factors) for finding stationary points in both deterministic and stochastic settings, the latter under sub-Gaussian noise. Central to our analysis is viewing Adafactor as an approximation of Adam, and the use of a new proxy step-size to approximate the unique adaptive step-size induced by Adafactor's matrix factorization and update clipping, along with an induction argument to control the gradient magnitude. Our findings may theoretically suggest that involving rank-1 matrix approximation of the second-moment matrix in Adam does not fundamentally hinder the convergence.

## 1 Introduction

Adaptive gradient-based methods, such as AdaGrad [12], RMSProp [41], Adadelta [47], Adam [22], and AMSGrad [37], among others, are efficient approaches in solving the following unconstrained stochastic optimization problem in deep learning fields:

$$\min_{\boldsymbol{X} \in \mathbb{R}^{n \times m}} f(\boldsymbol{X}) = \mathbb{E}_{\boldsymbol{Z} \in \mathcal{P}}[l(\boldsymbol{X}; \boldsymbol{Z})], \tag{1}$$

where $f$ is a smooth potentially non-convex function, $\mathcal{P}$ denotes a probability distribution and $\boldsymbol{X}$ denotes all the trainable weights of the model[2]. During the training process, these adaptive methods store the historical gradients' information to automatically tune their step-sizes. For example, both RMSProp and Adam maintain the exponential moving average of squared gradients, and AdaGrad stores the accumulation of squared gradients. Despite their effectiveness, adaptive gradient algorithms incur memory overhead compared to standard gradient descent, as they must store additional gradient statistics (e.g., first and second moments in Adam). This may become problematic when training large-scale models, such as GPT-3 [4], which contains over 175 billion parameters. The extra memory requirements may limit batch sizes or model complexity, posing challenges for resource-constrained training environments.

---

[*]The corresponding author is Junhong Lin.
[2]We consider the matrix parameter following the same setup in [38].

Adafactor [38] was proposed as a memory-efficient alternative to Adam, and subsequently many other memory-efficient optimization algorithms have been developed recently, see e.g., [38, 1, 31, 23, 32] and the references therein. Unlike Adam, which maintains per-parameter first and second moments of gradients, Adafactor employs a rank-1 matrix factorization to approximate the second-moment matrix. This reduces memory usage for the second-moment from $\mathcal{O}(mn)$ to $\mathcal{O}(m+n)$ with tracking only the exponential moving averages of the row and column sums of the squared gradient matrix. Additionally, Adafactor removes Adam's first-moment buffer and incorporates update clipping to improve training stability. In real applications, several LLMs including PaLM [8][3] and T5 [36] have adopted Adafactor as one of their main optimizers [53], and recent numerous studies on memory-efficient optimization algorithms have adopted Adafactor as the benchmark algorithm for comparative experiments.

The given empirical results reveal that Adafactor achieves comparable performance to RMSProp/Adam on training Transformer models [38], despite discarding part of the gradient information to save memory. Unlike Adam, whose convergence theory has been recently studied, e.g., [46, 55, 11, 50, 24, 42, 19], theoretical analysis for Adafactor remains absent to the best of our knowledge, though the algorithm was proposed several years ago. Specifically, it is unknown whether Adafactor can guarantee to find a stationary point as Adam for non-convex smooth optimization, and if so, what its specific convergence rate is and what conditions on hyper-parameters are required. We believe that the analysis is challenging, largely due to matrix factorization and update clipping mechanisms.

In this paper, we take the first step to analyze Adafactor's convergence theory for non-convex smooth optimization problems with unbounded gradients. Our main theoretical results are summarized as follows.

- With an appropriately chosen step-size and any decay rate $\beta_{2,k} \in [0,1)$, full-batch Adafactor can find a stationary point with a rate of $\mathcal{O}(1/T)$, matching that of Gradient Descent (GD) and the lower bound for first-order methods [5] up to constant factors.

- The stochastic Adafactor without update clipping can attain the convergence rate of $\tilde{\mathcal{O}}(1/\sqrt{T})$ under a common step-size parameter $\rho_k \sim \mathcal{O}(1/\sqrt{k})$ and a decay rate $\beta_{2,k} = 1 - 1/k$. The convergence rate is optimal up to logarithmic factors, matching the lower bound in [2].

- Adafactor with update clipping attains the nearly optimal convergence rate of $\tilde{\mathcal{O}}(1/\sqrt{T})$, provided that the clipping threshold and hyper-parameters are chosen appropriately.

We finally provide some simple numerical experiments on natural language processing to complement our theoretical results.

The analysis is non-trivial compared to memory-unconstrained adaptive methods such as AdaGrad and Adam due to the unique matrix factorization and update clipping. The core of our analysis is viewing Adafactor as an approximation of Adam, and designing a new proxy step-size to approximate the complicated adaptive step-size, while simultaneously breaking the correlation with stochastic gradients. In addition, we rely on an induction argument to prove that the objective function value is non-increasing in full-batch cases and that the gradient magnitude remains uniformly bounded during the training process in stochastic cases.

The rest of the paper is organized as follows. The next section briefly mentions some of the most relevant works. Section 3 presents some necessary notations and problem setups. Section 4 reviews Adafactor and its major differences to RMSProp/Adam. Sections 5 and 6 provide convergence bounds for full-batch Adafactor and stochastic Adafactor (without update clipping), respectively. Section 7 investigates Adafactor with the update clipping. Section 8 summarizes the main proof challenges and the proof novelty. Section 9 briefly presents experimental results to complement our theory. All the detailed proofs and some experiments can be found in the appendix.

## 2   Additional related work

We briefly list some typical works, due to page limitations.

---

[3]PaLM applies Adafactor without matrix factorization.

**Convergence of memory-unconstrained adaptive methods.** In the early stages, most works focus on the regret bound of adaptive methods on (online) convex optimization, e.g., [12, 40] for AdaGrad, [22, 37] for Adam and AMSGrad. Several works study the convergence of adaptive methods for non-convex smooth optimization, including [26, 44, 21, 13, 43, 3, 29] for AdaGrad-Norm, [43, 29, 20] for AdaGrad, [39, 25] for RMSProp, [54] for AMSGrad, and [46, 10, 55, 11, 6, 17, 50, 24, 42, 19, 7] for Adam. This body of work for non-convex smooth optimization consistently derives a convergence rate of $\tilde{\mathcal{O}}(1/\sqrt{T})$, with differences mainly on the noise and smooth assumptions, hyper-parameter dependencies and logarithmic factors in convergence bounds.

**Memory efficient algorithms.** The aforementioned memory-unconstrained adaptive methods such as AdaGrad and Adam require additional memory usage to store gradient-related statistics compared to traditional gradient descent methods. Consequently, a line of works focus on reducing the memory usage of such adaptive methods. For instance, [1] presents a variant of AdaGrad, called SM3, by maintaining $k$ sets of gradient accumulators. Both Adafactor and CAME [31] use matrix factorization to approximate the second moment of gradients in Adam. GaLore [51] factorizes the gradients through Singular Value Decomposition (SVD) before they enter the optimizer state. [32] proposes a variant of Adam called MicroAdam by compressing both gradients and error feedbacks. Adapprox [52] leverages randomized low-rank matrix approximation for Adam's second moment estimator. [23] develops a 4-bit Adam using quantization techniques to compress the first and second moment estimators in Adam. [49] reduces the memory by cutting down the learning rate resources in Adam.

However, most of these works provide empirical convergence results, with scarce exceptions on theoretical analysis. [1] establishes a regret bound in convex and bounded-stochastic-gradient setting for SM3. [32] provides convergence guarantees in expectation for MicroAdam with the assumptions of bounded gradients and well-behaved compression operators in non-convex smooth settings. Notably, these algorithms differ structurally from Adafactor, resulting in key differences in the proof. Moreover, our results hold with high probability without requiring bounded gradients or convexity assumptions.

Another line of works also use the idea of memory-efficiency over full-matrix preconditioned gradient methods. For example, works such as [18, 14, 45, 28], employ various techniques to approximate Hessian matrices in a memory-efficient way. [18] and [28] provide convergence bounds for their proposed algorithms in convex settings, assuming certain bounded gradient/Hessian-related terms.

**Notations.** For any positive integer $T$, let $[T] = \{1, 2, \cdots, T\}$. $\|\cdot\|_F$ and $\|\cdot\|_\infty$ denote the Frobenius norm and $\ell_\infty$-norm, respectively. $a \sim \mathcal{O}(b)$ and $a \leq \mathcal{O}(b)$ denote $a = C_0 b$ and $a \leq C_0 b$ for some positive constant $C_0$. For any two matrices $\boldsymbol{X} = (x_{ij})_{ij}, \boldsymbol{Y} = (y_{ij})_{ij} \in \mathbb{R}^{n \times m}$, we define $\langle \boldsymbol{X}, \boldsymbol{Y} \rangle = \sum_{i=1}^{n} \sum_{j=1}^{m} x_{ij} y_{ij}$. $\boldsymbol{X} \odot \boldsymbol{Y}$, $\frac{\boldsymbol{X}}{\boldsymbol{Y}}$ or $\boldsymbol{X}/\boldsymbol{Y}$, and $\sqrt{\boldsymbol{X}}$ denote the element-wise product, quotient, and square root, respectively. $\boldsymbol{0}_n$ and $\boldsymbol{1}_n$ denote the $n$-dimensional zero and one vectors respectively, and $\boldsymbol{1}_{n \times m}$ denotes the $n \times m$-dimensional matrix of ones. For any sequence $\{\alpha_i\}_{i \geq 1}$, we define $\sum_{i=a}^{b} \alpha_i = 0$ and $\prod_{i=a}^{b} \alpha_i = 1$ if $a > b$. $\chi_A$ denotes the indicator function with the set $A$. We define $\mathrm{RMS}(\boldsymbol{X}) = \sqrt{\frac{1}{mn} \sum_{i=1}^{n} \sum_{j=1}^{m} x_{ij}^2}$.

## 3 Problem setup

We consider unconstrained stochastic optimization in (1) over $\mathbb{R}^{n \times m}$ under the Frobenius norm. The objective function $f : \mathbb{R}^{n \times m} \to \mathbb{R}$ is differentiable. Given an $n \times m$ matrix $\boldsymbol{X}$, we assume a gradient oracle that returns a random matrix $g(\boldsymbol{X}, \boldsymbol{Z}) \in \mathbb{R}^{n \times m}$ dependent on the random sample $\boldsymbol{Z}$. The gradient of $f$ at $\boldsymbol{X}$ is denoted by $\nabla f(\boldsymbol{X}) \in \mathbb{R}^{n \times m}$.

**Assumptions.** We make the following assumptions throughout the paper.

- **(A1)** $L$-smoothness: for any $\boldsymbol{X}, \boldsymbol{Y} \in \mathbb{R}^{n \times m}$, $\|\nabla f(\boldsymbol{Y}) - \nabla f(\boldsymbol{X})\|_F \leq L\|\boldsymbol{Y} - \boldsymbol{X}\|_F$;
- **(A2)** Bounded below: there exists $f^* > -\infty$ such that $f(\boldsymbol{X}) \geq f^*, \forall \boldsymbol{X} \in \mathbb{R}^{n \times m}$;
- **(A3)** Unbiased estimator: the gradient oracle returns an unbiased estimator of $\nabla f(\boldsymbol{X})$, i.e., $\mathbb{E}[g(\boldsymbol{X}, \boldsymbol{Z}) \mid \boldsymbol{X}] = \nabla f(\boldsymbol{X}), \forall \boldsymbol{X} \in \mathbb{R}^{n \times m}$;
- **(A4)** Sub-Gaussian noise: for $\sigma > 0$, $\mathbb{E}\left[\exp\left(\frac{\|g(\boldsymbol{X}, \boldsymbol{Z}) - \nabla f(\boldsymbol{X})\|_F^2}{\sigma^2}\right) \Big| \boldsymbol{X}\right] \leq \mathrm{e}, \forall \boldsymbol{X} \in \mathbb{R}^{n \times m}$.

---
**Algorithm 1** Adafactor
---

**Input:** Horizon $T$, initialization $\boldsymbol{X}_1 \in \mathbb{R}^{n \times m}$, $\boldsymbol{R}_0 = \boldsymbol{0}_m$, $\boldsymbol{C}_0 = \boldsymbol{0}_n^\top$, step-size parameters $\{\rho_k\}_{k \geq 1}$, decay rates $\{\beta_{2,k}\}_{k \geq 1} \in [0,1)$, regularization constant $\epsilon_1 > 0$, clipping threshold $d$.
  **for** $k = 1, \cdots, T$ **do**
    Draw a random sample $\boldsymbol{Z}_k$ and $\boldsymbol{G}_k = g(\boldsymbol{X}_k, \boldsymbol{Z}_k)$;
    $\boldsymbol{R}_k = \beta_{2,k}\boldsymbol{R}_{k-1} + (1 - \beta_{2,k})(\boldsymbol{G}_k \odot \boldsymbol{G}_k + \epsilon_1 \boldsymbol{1}_n \boldsymbol{1}_m^\top)\boldsymbol{1}_m$;
    $\boldsymbol{C}_k = \beta_{2,k}\boldsymbol{C}_{k-1} + (1 - \beta_{2,k})\boldsymbol{1}_n^\top(\boldsymbol{G}_k \odot \boldsymbol{G}_k + \epsilon_1 \boldsymbol{1}_n \boldsymbol{1}_m^\top)$;
    $\boldsymbol{W}_k = (\boldsymbol{R}_k\boldsymbol{C}_k)/(\boldsymbol{1}_n^\top \boldsymbol{R}_k)$;
    $\boldsymbol{U}_k = \boldsymbol{G}_k/\sqrt{\boldsymbol{W}_k}$;
    $\eta_k = \rho_k/\max\{1, \mathrm{RMS}(\boldsymbol{U}_k)/d\}$;
    $\boldsymbol{X}_{k+1} = \boldsymbol{X}_k - \eta_k \cdot \boldsymbol{G}_k/\sqrt{\boldsymbol{W}_k}$;
  **end for**

---

Assumptions (A1)–(A4) are standard in the convergence analysis for smooth non-convex optimization. In particular, the sub-Gaussian noise assumption is widely used in the convergence analysis of gradient-based methods, including SGD [15], AdaGrad [27, 21, 29], and Adam [24].

## 4 A review of Adafactor

In this section, we briefly introduce Adafactor and highlight its major differences from Adam. The pseudocode for Adafactor is presented in Algorithm 1.

**Matrix factorization.** Throughout the training process, Adam maintains two $n \times m$ matrices, $\boldsymbol{M}_k$ and $\boldsymbol{V}_k$, using the exponential moving average update: for $\beta_{1,k}, \beta_{2,k} \in [0,1)$,

$$\boldsymbol{M}_k = \beta_{1,k}\boldsymbol{M}_{k-1} + (1 - \beta_{1,k})\boldsymbol{G}_k, \quad \boldsymbol{V}_k = \beta_{2,k}\boldsymbol{V}_{k-1} + (1 - \beta_{2,k})(\boldsymbol{G}_k \odot \boldsymbol{G}_k), \tag{2}$$

which results in tripled memory usage. The key innovation of Adafactor in improving memory usage is to approximate $\boldsymbol{V}_k$ as the outer product of two rank-1 matrices $\boldsymbol{R}_k$ and $\boldsymbol{C}_k/(\boldsymbol{1}_n^\top \boldsymbol{R}_k)$, as shown in Algorithm 1. Moreover, $\boldsymbol{R}_k$ and $\boldsymbol{C}_k$ are exactly the row sums and column sums of $\boldsymbol{V}_k$, and they also follow the exponential moving average update. Therefore, Adafactor only maintains two rank-1 matrices $\boldsymbol{R}_k$ and $\boldsymbol{C}_k$, significantly reducing the memory usage of storing $\boldsymbol{V}_k$ from $\mathcal{O}(mn)$ to $\mathcal{O}(m+n)$.

**Increasing decay rate.** In Adam, corrective terms are introduced into $\boldsymbol{M}_k$ and $\boldsymbol{V}_k$, leading to two decay rates that increase toward one. Theoretically, it has been demonstrated that a value close to one for $\beta_{2,k}$ would ensure the convergence, e.g., [11, 55, 50] whereas a constant one may lead to divergence [37]. Inspired by this observation, Adafactor uses an increasing second-moment decay rate $\beta_{2,k} = 1 - 1/k^c, c > 0$ to replace corrective terms. As pointed out by [38], this setting allows for enjoying the stability of a low $\beta_{2,k}$ at the early stages of training and the insurance of convergence from a high $\beta_{2,k}$ as the run progresses. Moreover, it leverages the bias correction.

**Update clipping.** Adafactor modifies the update process by discarding the first-order moment $\boldsymbol{M}_k$ and instead applies an update clipping technique inside the step-size $\eta_k$. It is worth highlighting that the update clipping involves dividing the root-mean-square of $\boldsymbol{U}_k$ when it exceeds a threshold $d$, which differs from the standard gradient-clipping with the form $\eta_k = \rho_k/\max\{1, \|\boldsymbol{G}_k\|_F/d\}$. This mechanism helps to calibrate the second-moment estimator $\boldsymbol{W}_k$ when it's larger-than-desired $\boldsymbol{G}_k \odot \boldsymbol{G}_k$. Empirical findings in [38] indicate that implementing update clipping leads to significant performance improvements when the learning-rate warm-up is not used.

## 5 Convergence bound for full-batch Adafactor

We first provide the convergence bound for the full-batch Adafactor. At each iteration, full-batch Adafactor obtains the gradient $\nabla f(\boldsymbol{X}_k)$ and then updates $\boldsymbol{R}_k, \boldsymbol{C}_k$ using $\nabla f(\boldsymbol{X}_k)$ instead of $\boldsymbol{G}_k$ in Algorithm 1. The proof can be found in Appendix A.

**Theorem 5.1.** *Let $\{\boldsymbol{X}_k\}_{k\geq 1}$ be generated by Algorithm 2, and Assumptions (A1) and (A2) hold. For any constants $c_0, d > 0$ and $\beta_{2,1} \in [0,1)$, we define*

$$G := \sqrt{2L(f(\boldsymbol{X}_1) - f^*)} + c_0, \quad \Delta := \max\{1, G^2\} + \frac{c_0}{d(1 - \beta_{2,1})}. \tag{3}$$

*If $0 \leq \beta_{2,k} < 1, \rho_k = \rho_0, \forall k \geq 1$ and*

$$\epsilon_1 = \frac{c_0}{dmn(1 - \beta_{2,1})}, \quad 0 < \rho_0 \leq \frac{c_0^3}{Ld^2 mnG\Delta^2}, \tag{4}$$

*then, for any $T \geq 1$,*

$$\min_{k\in[T]} \|\bar{\boldsymbol{G}}_k\|_F^2 \leq \frac{2G\Delta\left(f(\boldsymbol{X}_1) - f^*\right)}{\rho_0 T}.$$

The result indicates that full-batch Adafactor can find a stationary point at a rate of $\mathcal{O}(1/T)$, matching that of Gradient Descent and the lower bound for deterministic non-convex smooth optimization [5] up to constant factors. We require $\epsilon_1 \sim \mathcal{O}\left(\frac{1}{mn}\right)$ and $\rho_0 \leq \mathcal{O}\left(\frac{1}{mn}\right)$. The setting for $\beta_{2,k}$ is mild, including the default setup in [38] where $\beta_{2,k} = 1 - 1/k^{0.8}$. In addition, we can set $\rho_0 \sim \mathcal{O}\left(\frac{1}{mn}\right)$ to derive a convergence bound of $\mathcal{O}(mn)$ with respect to the dimension.

## 6 Stochastic Adafactor without update clipping

In the stochastic case, we start from the simple scenario where $\eta_k = \rho_k$, dropping the update clipping $1/\max\{1, \text{RMS}(\boldsymbol{U}_k)/d\}$. The main reasons are as follows.

- As a first step toward theoretically investigating the convergence of Adafactor, we retain its most essential component—the matrix factorization—while temporarily omitting the relatively secondary update clipping. This simplification makes the proof more tractable.

- As pointed out in the experiments from [38], Adafactor's performance shows little difference with and without update clipping when implementing learning rate warm-up which is a popular method in deep learning [53].

We now present the probabilistic convergence bound for Adafactor without update clipping as follows. The detailed proof can be found in Appendix B.

**Theorem 6.1.** *Let $\{\boldsymbol{X}_k\}_{k\geq 1}$ be generated by Algorithm 1 with $\eta_k = \rho_k, \forall k \geq 1$ and Assumptions (A1)-(A4) hold. For any $T \geq 1, \delta \in (0, 1/2), \lambda_0, c_0 > 0$, we define*

$$H^2 := 2L(f(\boldsymbol{X}_1) - f^*) + \frac{12\sigma^2\lambda_0}{c_0}\log\left(\frac{T}{\delta}\right) + \frac{4\lambda_0(24 + \lambda_0)(1 + \log T)}{c_0^2},$$

$$\Sigma_H := H + \sigma\sqrt{\log\left(\frac{\mathrm{e}T}{\delta}\right)}, \quad \mathcal{H} := \Sigma_H^2 + c_0\sqrt{mn}. \tag{5}$$

*If $\rho_0$ satisfies that*

$$0 < \rho_0 \leq \frac{\lambda_0}{L}\min\left\{\frac{1}{\sqrt{\mathcal{H}}}, \frac{1}{\Sigma_H^2 \mathcal{H}^{3/2}}, \frac{1}{\Sigma_H\sqrt{\mathcal{H}}}\right\}, \tag{6}$$

*and other parameters satisfy that $\epsilon_1 = \frac{c_0}{\sqrt{mn}}, \beta_{2,1} = \frac{1}{2}, \rho_1 = \rho_0$, and for some constant $c \in [0, 1]$,*

$$\beta_{2,k} = 1 - \frac{1}{k^c}, \quad \rho_k = \frac{\rho_0}{k^{1-c/2}}, \quad \forall k \geq 2, \tag{7}$$

*then, with probability at least $1 - 2\delta$,*

$$\min_{k\in[T]} \|\nabla f(\boldsymbol{X}_k)\|_F^2 \leq \frac{H^2}{\rho_0 LT^{c/2}}\left(H + \sigma\sqrt{\log\left(\frac{\mathrm{e}T}{\delta}\right)} + \sqrt{c_0}\right). \tag{8}$$

**Convergence rate.** Since $H^2 \sim \mathcal{O}(\log T)$, we can set $\rho_0 \approx \frac{\lambda_0}{L\Sigma_H^2 \mathcal{H}^{3/2}} \sim \mathcal{O}\left(\frac{1}{\log^{5/2}(T)}\right)$ satisfying (6), which leads to $\mathcal{O}\left(\frac{\log^4(T)}{T^{c/2}}\right)$ order of convergence rate. With logarithmic factors ignored, Adafactor can achieve the nearly optimal $\tilde{\mathcal{O}}(1/\sqrt{T})$ convergence rate when $c = 1$, matching the ones for RMSProp/Adam in literature and the lower bound [2] for stochastic non-convex smooth optimization.

**Hyper-parameter setups.** Our result indicates that the optimal rate is attained with $\beta_{2,k} = 1 - 1/k, \rho_k = \rho_0/\sqrt{k}$, a pattern commonly appeared in theoretical analyses of RMSProp [55, 25] and Adam [55]. When $c$ increases from 0 to 1, the convergence rate also improves. We also test our hyperparameter setup empirically, indicating a similar improvement as $c$ increases, see Figure 1 and Table 1 in the appendix.

We apply polynomial decay step-size parameters, which have been widely used in existing literature such as [33]. We also require $\rho_0 \leq \mathcal{O}\left(\frac{1}{\text{poly}(\log T)}\right)$ and $\epsilon_1 \sim \mathcal{O}\left(\frac{1}{\sqrt{mn}}\right)$.

**Dimension dependency.** We can set $\rho_0 \sim \mathcal{O}(\mathcal{H}^{-3/2}) \sim \mathcal{O}((mn)^{-3/4})$ given that $H^2 \sim \mathcal{O}(1)$ and $\mathcal{H} \sim \mathcal{O}(\sqrt{mn})$ in terms of dimension dependency. With the setup, the convergence bound is $\mathcal{O}((mn)^{3/4})$ with respect to the dimension. Under the assumptions of smoothness, [29, 20] derive bounds of at least $\mathcal{O}(mn)$ with respect to the dimension for AdaGrad. For Adam and RMSProp, many existing works [11, 50, 42, 25] derive $\mathcal{O}(\text{poly}(mn))$ dependency while [24] derive a dimension-free convergence bound. Our convergence bounds show comparable dimension dependency to most results for AdaGrad and Adam, though a gap remains toward achieving fully dimension-free guarantees, and improving the dimension dependency could be further investigated in the future.

**Time-invariant $\beta_{2,k}$.** The following convergence bound sets a time-invariant $\beta_{2,k} = 1 - 1/T, \forall k \in [T]$, a setting commonly used in Adam's convergence results [11, 42, 19]. The result indicates that Adafactor can still achieve $\tilde{O}(1/\sqrt{T})$ convergence rate. The detailed proof is in Appendix B.5.

**Corollary 1.** *Let $\{\boldsymbol{X}_k\}_{k\geq 1}$ be generated by Algorithm 1 with $\eta_k = \rho_k, \forall k \geq 1$ and Assumptions (A1)-(A4) hold. Let $T \geq 1, \delta \in (0, 1/2)$, $H$ and $\mathcal{H}$ be defined in (5). If $\beta_{2,1} = \frac{1}{2}, \beta_{2,k} = 1 - \frac{1}{T}, \forall k \in [T] \setminus \{1\}, \rho_k = \frac{\rho_0}{\sqrt{T}}, \forall k \in [T], \epsilon_1 = \frac{c_0}{\sqrt{mn}}$, and $\rho_0 \leq \frac{\lambda_0}{L} \min\left\{\frac{1}{\sqrt{\mathcal{H}}}, \frac{1}{2\Sigma_H^2 \mathcal{H}^{3/2}}, \frac{1}{\Sigma_H \sqrt{\mathcal{H}}}\right\}$, then it holds that with probability at least $1 - 2\delta$,*

$$\frac{1}{T} \sum_{k=1}^{T} \|\nabla f(\boldsymbol{X}_k)\|_F^2 \leq \frac{H^2}{\rho_0 L\sqrt{T}} \left( H + \sigma\sqrt{\log\left(\frac{\mathrm{e}T}{\delta}\right)} + \sqrt{c_0} \right).$$

# 7 Stochastic Adafactor with update clipping

In this section, we consider the update clipping and slightly change the threshold $d$ in Algorithm 1 to a time-varying threshold $d_k$. The update clipping in Adafactor differs from the standard clipping mechanism, bringing some more essential challenges for analysis. In what follows, we demonstrate that incorporating such clipping can still ensure convergence for Adafactor under sub-Gaussian noise. The detailed proof is in Appendix C.

**Theorem 7.1.** *Let $\{\boldsymbol{X}_k\}_{k\geq 1}$ be generated by Algorithm 1 with $d$ replaced by $d_k$ for any $k$-th iteration. Let Assumptions (A1)-(A4) hold. For any $T \geq 1, \delta \in (0, 1/2), \lambda_0, c_0 > 0$ and $\alpha > 1$, let*

$$I^2 := 2L(f(\boldsymbol{X}_1) - f^*) + \frac{4\lambda_0(24 + \lambda_0)(1 + \log T)}{c_0^2} + \frac{4\sqrt{\delta}(1 + \log T)}{c_0}$$

$$+ \frac{192\lambda_0}{c_0} \log\left(\frac{T}{\delta}\right) + \frac{2^{\alpha+1}\lambda_0(1 + \log T)}{(mn)^{(\alpha-1)/2}c_0^\alpha}. \tag{9}$$

*Also, let $\Sigma_I := I + \sigma\sqrt{\log\left(\frac{\mathrm{e}T}{\delta}\right)}$ and $\mathcal{I} := \Sigma_I^2 + c_0\sqrt{mn}$. If $\rho_0$ satisfies that*

$$0 < \rho_0 \leq \frac{\lambda_0}{L} \min\left\{\frac{1}{\Sigma_I^2\sqrt{\mathcal{I}}}, \frac{1}{\Sigma_I^2\mathcal{I}^{3/2}}, \frac{1}{\Sigma_I\sqrt{\mathcal{I}}}, \frac{1}{I(\Sigma_I\sqrt{\mathcal{I}})^\alpha}\right\}, \tag{10}$$

$\epsilon_1, \beta_{2,k}$ and $\rho_k$ follow the setups in (7) for any $c \in [0,1]$, and $d_k \geq k^{\frac{c}{2(\alpha-1)}}, \forall k \in [T]$, then, with probability at least $1 - 2\delta$,

$$\min_{k \in [T]} \|\bar{\boldsymbol{G}}_k\|_F^2 \leq \frac{I^2}{\rho_0 L T^{c/2}} \left( I + \sigma \sqrt{\log\left(\frac{\mathrm{e}T}{\delta}\right)} + \sqrt{c_0} \right).$$

**Convergence rate.** With $I^2 \sim \mathcal{O}(\log T)$, both $\Sigma_I^2$ and $\mathcal{I}$ are $\mathcal{O}(\log T)$ order and the typical setup of $\rho_0$ is $\mathcal{O}\left(1/\log^{\max\{\frac{5}{2}, \frac{1+2\alpha}{2}\}}(T)\right)$ satisfying (10), which leads to $\mathcal{O}\left(\frac{\log^{\max\{4, 2+\alpha\}}(T)}{T^{c/2}}\right)$ order for the convergence bound. When $c = 1$, Adafactor still achieves the nearly optimal $\tilde{\mathcal{O}}(1/\sqrt{T})$ rate. In addition, we can set $\rho_0 \sim \mathcal{O}(\mathcal{I}^{-3/2}) \sim \mathcal{O}((mn)^{-3/4})$ given that $I^2 \sim \mathcal{O}(1)$ and $\mathcal{I} \sim \mathcal{O}(\sqrt{mn})$ with respect to the dimension. Under this setup, the convergence bound is $\mathcal{O}((mn)^{3/4})$ with respect to the dimension.

**Impact of update clipping.** When incorporating update clipping, $\alpha$ influences the selection of $\rho_0$ and $d_k$, and the $\log T$ order in the convergence bound. The results suggest that the update clipping does not significantly impact the convergence rate under sub-Gaussian noise. We hypothesize that under sub-Gaussian (light-tailed) noise, update clipping is not necessary for ensuring convergence. However, for other cases such as the heavy-tailed noise, update clipping may play a crucial role, similar to the role of standard gradient clipping, as demonstrated in e.g., [9, 48, 16, 7].

We require $d_k$ to increase with steps $k$. At the early stages of training where the updates are usually unstable [38, Figure 1], $d_k$ is small to ensure the clipping works effectively. As training progresses, the sequences become more stable. Consequently, there is less need for update clipping, corresponding to a relatively large $d_k$. We test this setup through some experiments, showing its comparable performance with the standard setting $d_k = 1$, see Figure 4 and Table 2 in the appendix.

**Time-invariant $\beta_{2,k}$.** We also provide the convergence bound with $\beta_{2,k} = 1 - 1/T$, which shares a similar form to the one in Corollary 1. The detailed proof is in Appendix C.4.

**Corollary 2.** *Let* $T \geq 1, \delta \in (0, 1/2)$, $I$ *and* $\mathcal{I}$ *be defined in Theorem 7.1. If* $\beta_{2,1} = \frac{1}{2}, \beta_{2,k} = 1 - \frac{1}{T}, \forall k \in [T] \setminus \{1\}, \rho_k = \frac{\rho_0}{\sqrt{T}}, \forall k \in [T], \epsilon_1 = \frac{c_0}{\sqrt{mn}}, d_k \geq k^{\frac{c}{2(\alpha-1)}}, \forall k \in [T]$ *and* $\rho_0 \leq \frac{\lambda_0}{L} \min\left\{\frac{1}{\Sigma_I^2 \sqrt{\mathcal{I}}}, \frac{1}{2\Sigma_I^2 \mathcal{I}^{3/2}}, \frac{1}{\Sigma_I \sqrt{\mathcal{I}}}, \frac{1}{I(\Sigma_I \sqrt{\mathcal{I}})^\alpha}\right\}$, *then it holds that with probability at least* $1 - 2\delta$,

$$\frac{1}{T} \sum_{k=1}^{T} \|\nabla f(\boldsymbol{X}_k)\|_F^2 \leq \frac{I^2}{\rho_0 L T^{c/2}} \left( I + \sigma \sqrt{\log\left(\frac{\mathrm{e}T}{\delta}\right)} + \sqrt{c_0} \right).$$

# 8 Summary of proof challenges and techniques

In this section, we will summarize the main proof challenges brought by Adafactor, which are essentially different from other memory-unconstrained adaptive methods such as Adam due to the unique matrix factorization and update clipping.

We let $\bar{\boldsymbol{G}}_k := \nabla f(\boldsymbol{X}_k)$ and begin by the descent lemma of the smoothness [34, Theorem 2.1.5],

$$f(\boldsymbol{X}_{k+1}) \leq f(\boldsymbol{X}_k) \underbrace{-\eta_k \left\langle \bar{\boldsymbol{G}}_k, \frac{\boldsymbol{G}_k}{\sqrt{\boldsymbol{W}_k}} \right\rangle}_{\textbf{(I)}} + \underbrace{\frac{L\eta_k^2}{2} \left\| \frac{\boldsymbol{G}_k}{\sqrt{\boldsymbol{W}_k}} \right\|_F^2}_{\textbf{(II)}}, \quad \forall k \geq 1. \tag{11}$$

Then, the following challenges arise from estimating **(I)** and **(II)**.

**Challenge I. Correlation between $\boldsymbol{G}_k$ and $\boldsymbol{W}_k$.** The classical method for estimating **(I)** is to decompose it as the "descent term" plus the "noise variance term":

$$\textbf{(I)} = \underbrace{-\eta_k \left\| \frac{\bar{\boldsymbol{G}}_k}{\sqrt[4]{\boldsymbol{W}_k}} \right\|_F^2}_{\text{descent term}} \underbrace{-\eta_k \left\langle \bar{\boldsymbol{G}}_k, \frac{\boldsymbol{G}_k - \bar{\boldsymbol{G}}_k}{\sqrt{\boldsymbol{W}_k}} \right\rangle}_{\text{noise variance}}.$$

For non-adaptive methods such as SGD, "noise variance" is a martingale difference sequence. However, its conditional expectation is not necessarily zero, and the property of martingale can no longer be used due to the correlation of $\boldsymbol{G}_k$ and $\boldsymbol{W}_k$ in Adafactor. Other adaptive methods such as AdaGrad and Adam, also face a similar problem. To overcome this, existing works for AdaGrad and Adam such as [44, 11, 42, 19] typically introduce a proxy step-size matrix $\boldsymbol{A}_k$ that is conditionally independent of $\boldsymbol{G}_k$ and decompose **(I)** as

$$(\mathbf{I}) = -\eta_k \left\| \frac{\bar{\boldsymbol{G}}_k}{\sqrt[4]{\boldsymbol{A}_k}} \right\|_F^2 - \underbrace{\eta_k \left\langle \bar{\boldsymbol{G}}_k, \frac{\boldsymbol{G}_k - \bar{\boldsymbol{G}}_k}{\sqrt{\boldsymbol{A}_k}} \right\rangle}_{\text{martingale difference}} + \underbrace{\eta_k \left\langle \bar{\boldsymbol{G}}_k, \boldsymbol{G}_k \odot \left( \frac{1}{\sqrt{\boldsymbol{A}_k}} - \frac{1}{\sqrt{\boldsymbol{W}_k}} \right) \right\rangle}_{\text{error}}. \quad (12)$$

For these works, proxy step-sizes are designed based on the linear update of $\boldsymbol{W}_k$, the adaptive part in step-sizes, such as (2). However, Adafactor uses a more complicated adaptive step-size with a non-linear update rule between $\boldsymbol{W}_k$ and $\boldsymbol{W}_{k-1}$, as shown in Algorithm 1, making existing proxy step-sizes not applicable.

**Solution.** We first define some temporary bounds for (stochastic) gradients: for fixed horizon $T$ and any $k \in [T]$, $D_k := \max_{s \in [k]} \|\bar{\boldsymbol{G}}_s\|_F, \Sigma_k := D_k + \sigma \sqrt{\log\left(\frac{eT}{\delta}\right)}$ and

$$\mathcal{G}_{k,1} := \Sigma_k^2 + m\epsilon_1, \quad \mathcal{G}_{k,2} := \Sigma_k^2 + n\epsilon_1, \quad \mathcal{G}_k := \Sigma_k^2 + mn\epsilon_1. \quad (13)$$

Relying on the property of sub-Gaussian noise, we can verify the following inequalities with probability at least $1 - \delta$ (an equivalent form of (42)),

$$\max_{s \in [T]} \|\boldsymbol{G}_s - \bar{\boldsymbol{G}}_s\|_F \le \sigma \sqrt{\log\left(\frac{eT}{\delta}\right)}, \quad \max_{s \in [k]} \|\boldsymbol{G}_s\|_F \le \Sigma_k, \quad \forall k \in [T]. \quad (14)$$

We design a new proxy step-size matrix $\boldsymbol{A}_k$ as follows:

$$\boldsymbol{A}_k := \frac{(\beta_{2,k}\boldsymbol{R}_{k-1} + (1 - \beta_{2,k})\mathcal{G}_{k,1} \cdot \mathbf{1}_n)(\beta_{2,k}\boldsymbol{C}_{k-1} + (1 - \beta_{2,k})\mathcal{G}_{k,2} \cdot \mathbf{1}_m^\top)}{\beta_{2,k}S_{k-1} + (1 - \beta_{2,k})\mathcal{G}_k}.$$

$\boldsymbol{A}_k$ satisfies two important properties: (a). It's conditionally independent with $\boldsymbol{G}_k - \bar{\boldsymbol{G}}_k$. Thereby, "martingale difference" term in (12) can be bounded through the concentration inequality. (b). The following "distance" between $\boldsymbol{W}_k$ and $\boldsymbol{A}_k$ can be estimated by $D_k$ multiplying a small term $\sqrt{1 - \beta_{2,k}}$ as $\beta_{2,k}$ is set to close enough to one: let $\boldsymbol{W}_k = (w_{ij}^{(k)})_{ij}, \boldsymbol{A}_k = (a_{ij}^{(k)})_{ij}$, then

$$\frac{\left| w_{ij}^{(k)} - a_{ij}^{(k)} \right|}{\sqrt{a_{ij}^{(k)}}} \le \mathcal{O}\left(D_k \sqrt{1 - \beta_{2,k}}\right), \quad \forall k \in [T], i \in [n], j \in [m].$$

Relying on this bound and the setups of $\eta_k$ and $\beta_{2,k}$ in (7), and probability event in (14), we get that

$$\sum_{k=1}^{t} \text{error} \le \sum_{k=1}^{t} \frac{\eta_k}{4} \left\| \frac{\bar{\boldsymbol{G}}_k}{\sqrt[4]{\boldsymbol{A}_k}} \right\|_F^2 + \mathcal{O}\left(\rho_0 \Sigma_t^2 \mathcal{G}_t^{3/2} \log t\right), \quad \forall t \in [T]. \quad (15)$$

We also refer to the proof of Proposition B.1 in the appendix for more details.

**Challenge II. Additional update clipping.** The first solution only considers the case where the update clipping is omitted. The update clipping introduces an even more complex adaptive step-size. We incorporate the new proxy step-size method in **Solution 1** and some techniques from the analysis of algorithms with standard clipping [9, 30, 35].

**Solution.** We first rewrite the update rule as

$$\boldsymbol{X}_{k+1} = \boldsymbol{X}_k - \rho_k \frac{\tilde{\boldsymbol{G}}_k}{\sqrt{\boldsymbol{W}_k}}, \quad \tilde{\boldsymbol{G}}_k = \frac{\boldsymbol{G}_k}{\max\{1, \text{RMS}(\boldsymbol{U}_k)/d_k\}}.$$

Then, we follow the design of $\boldsymbol{A}_k$ in the first solution and provide a decomposition for $(\mathbf{I})$ in (11),

$$
(\mathbf{I}) = \underbrace{-\rho_k \left\| \frac{\bar{\boldsymbol{G}}_k}{\sqrt[4]{\boldsymbol{A}_k}} \right\|_F^2}_{\text{descent term}} + \underbrace{\rho_k \left\langle \bar{\boldsymbol{G}}_k, \left( \frac{1}{\sqrt{\boldsymbol{A}_k}} - \frac{1}{\sqrt{\boldsymbol{W}_k}} \right) \odot \tilde{\boldsymbol{G}}_k \right\rangle}_{\text{error 1}}
$$

$$
\underbrace{-\rho_k \left\langle \bar{\boldsymbol{G}}_k, \frac{\tilde{\boldsymbol{G}}_k}{\sqrt{\boldsymbol{A}_k}} - \mathbb{E}_{\boldsymbol{Z}_k} \left[ \frac{\tilde{\boldsymbol{G}}_k}{\sqrt{\boldsymbol{A}_k}} \right] \right\rangle}_{\text{martingale difference}} + \underbrace{\rho_k \left\langle \bar{\boldsymbol{G}}_k, \frac{\bar{\boldsymbol{G}}_k}{\sqrt{\boldsymbol{A}_k}} - \mathbb{E}_{\boldsymbol{Z}_k} \left[ \frac{\tilde{\boldsymbol{G}}_k}{\sqrt{\boldsymbol{A}_k}} \right] \right\rangle}_{\text{error 2}},
$$

where $\boldsymbol{Z}_k$ is the $k$-th random sample. Note that "error 1" shares a similar form as "error" in (12), which can be estimated similarly as in (15). The critical point is to handle the additional "error 2". With $\boldsymbol{A}_k$ conditionally independent with $\boldsymbol{Z}_k$ and $\bar{\boldsymbol{G}}_k = \mathbb{E}_{\boldsymbol{Z}_k}[\boldsymbol{G}_k]$ from Assumption 3,

$$
\text{error } 2 \le \rho_k \left\| \frac{\bar{\boldsymbol{G}}_k}{\sqrt{\boldsymbol{A}_k}} \right\|_F \cdot \mathbb{E}_{\boldsymbol{Z}_k} \|\boldsymbol{\Omega}_k\|_F, \quad \boldsymbol{\Omega}_k := \boldsymbol{G}_k \left( 1 - \frac{1}{\max\{1, \|\boldsymbol{U}_k\|_F/(d_k\sqrt{mn})\}} \right). \quad (16)
$$

Under the probability event of (14), we will estimate $\mathbb{E}_{\boldsymbol{Z}_k} \|\boldsymbol{\Omega}_k\|_F$ which is solely dependent on $\boldsymbol{Z}_1, \cdots, \boldsymbol{Z}_{k-1}$. Then, we can further derive that

$$
\mathbb{E}_{\boldsymbol{Z}_k} \|\boldsymbol{\Omega}_k\|_F \le \Sigma_k \sqrt{\frac{\delta}{T}} + \Sigma_k^{\alpha} \left( \frac{2\sqrt{\mathcal{G}_k}}{d_k m n \epsilon_1} \right)^{\alpha-1}. \quad (17)
$$

Combining the above, and applying setups for $d_k, \rho_k$ and $\epsilon_1$, we get the following bound under (14),

$$
\sum_{k=1}^t \text{error } 2 \le \mathcal{O} \left( \sum_{k=1}^t \frac{\rho_0 D_k \left( \Sigma_k \sqrt{\mathcal{G}_k} \right)^{\alpha}}{k} \right) \le \mathcal{O} \left( \rho_0 D_t \left( \Sigma_t \sqrt{\mathcal{G}_t} \right)^{\alpha} \log t \right), \quad \forall t \in [T].
$$

For more details, we refer to the proof of Proposition C.1 in the appendix.

**Challenge III. Potential unbounded gradient magnitude.** Throughout the paper, we do not assume the gradient magnitude is bounded. Therefore, we can only estimate $(\mathbf{I})$ and $(\mathbf{II})$ through the temporary bounds $D_k, \Sigma_k$ and $\mathcal{G}_k$ in (13).

**Solution (stochastic case).** First, based on the estimations for $(\mathbf{I})$ and $(\mathbf{II})$, one can derive that for some increasing positive function $\phi(x)$, with probability at least $1 - \delta$,

$$
f(\boldsymbol{X}_{t+1}) - f^* \le -\frac{1}{2} \sum_{k=1}^t \eta_k \left\| \frac{\bar{\boldsymbol{G}}_k}{\sqrt[4]{\boldsymbol{A}_k}} \right\|_F^2 + \mathcal{O} \left( \rho_0 \phi(D_t) \log \left( \frac{T}{\delta} \right) \right), \quad \forall t \in [T]. \quad (18)
$$

Then, we use an induction argument to restrict the gradient magnitude. The induction will start by verifying $D_1 \le H$ and then assume that $D_t \le H$ for some $t \in [T]$ where $H$ is a value defined with $\mathcal{O}(\sqrt{\log(T/\delta)})$ order in prior. Using the induction assumption and $\|\bar{\boldsymbol{G}}_{t+1}\|_F^2 \le 2L(f(\boldsymbol{X}_{t+1}) - f^*)$ into (18),

$$
\|\bar{\boldsymbol{G}}_{t+1}\|_F^2 \le -L \sum_{k=1}^t \eta_k \left\| \frac{\bar{\boldsymbol{G}}_k}{\sqrt[4]{\boldsymbol{A}_k}} \right\|_F^2 + \mathcal{O} \left( \rho_0 L \phi(H) \log \left( \frac{T}{\delta} \right) \right) \le \mathcal{O} \left( c_0 \log \left( \frac{T}{\delta} \right) \right), \quad (19)
$$

where the last inequality applies the setup $\rho_0 \le \frac{c_0}{L\phi(H)}$. Then, we derive that $\|\bar{\boldsymbol{G}}_{t+1}\|_F^2 \le H^2$ from (19) as $H^2$ is $\mathcal{O}(\log(T/\delta))$ order and can be set equal to the RHS of (19). The induction is thereby complete, and the gradient magnitude is bounded by $H$. We refer to the proof of Proposition B.1 for more details.

**Solution (full-batch case).** In the noiseless case, $(\mathbf{I})$ and $(\mathbf{II})$ can be cancelled with each other through a proper selection of $\eta_k$. Relying on this, we can use an induction to derive a stronger result where $f(\boldsymbol{X}_t)$ is non-increasing with $t$. See the proof of Proposition A.1 for more details.

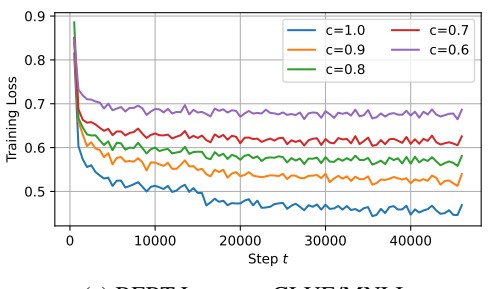
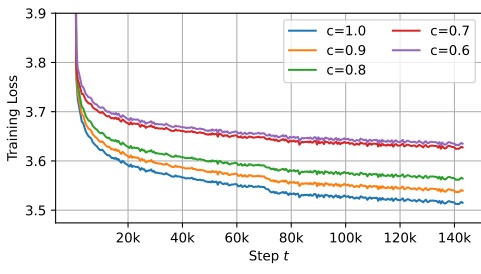

(a) BERT-Large on GLUE/MNLI          (b) GPT-2 on BookCorpus dataset

Figure 1: Training loss vs steps for different decay rates using Adafactor (no update clipping)

## 9   Experiment

Many existing works, such as [38, 51, 32, 53], have empirically demonstrated the convergence of Adafactor, showing that it achieves comparable performance to Adam in training NLP models.

While our main contribution lies in theoretical parts, we also test our hyper-parameter setups in the full fine-tuning (FFT) scenario. We train BERT-Base and BERT-Large on GLUE/MNLI and GPT-2 on BookCorpus dataset. We follow the setup in Theorem 6.1 and require $c$ to range from $0.6$ to $1.0$. Training loss curves are presented in Figure 1 and Figure 3, and test accuracy is reported in Table 1 in the appendix. The results show that as $c$ increases, both the training loss and the test accuracy improve, complementing our theoretical findings. The detailed training settings can be found in Appendix D.1.

We also compare our configuration at $c = 1$ (the optimal selection in theoretical) with the default setting proposed in [38] and with Adam, finding that their performances remain comparable. When incorporating update clipping, we test the increasing clipping threshold $d_k = k^{\frac{c}{2(\alpha-1)}}$ proposed in Theorem 7.1, and find its performance to be comparable to the default setting where $d_k = 1$ and to Adam. Detailed experimental results are provided in Appendix D.2.

## 10   Conclusion

In this paper, we take the first step toward understanding the convergence of Adafactor in the non-convex smooth landscape under sub-Gaussian noise. Our theoretical results indicate that with the proper hyper-parameter setups, Adafactor can achieve the nearly optimal convergence rate, matching the lower bound for first-order methods in full-batch cases up to constant factors, and stochastic cases up to logarithmic factors.

**Limitations.**    First, the convergence behavior of Adafactor with a constant clipping threshold, which may be more common in practical applications, remains theoretically unexplored. Second, it remains unknown whether Adafactor can still converge under other noise assumptions, such as heavy-tail noise and affine variance noise. Third, the convergence results for Adafactor are established under the standard smoothness assumption. It would be interesting to further investigate the convergence under more general smoothness conditions that better reflect practical applications, such as $(L_0, L_1)$-smoothness. Finally, it's beneficial to further support our theoretical results through experiments on large language models.

## Acknowledgement

This work was supported in part by the NSFC under grant number 12471096, and the National Key Research and Development Program of China under grant number 2021YFA1003500.

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

# A  Proof detail for Theorem 5.1

We first provide the form of full-batch Adafactor as follows. The only difference to Algorithm 1 is the replacement of the stochastic gradient by the gradient $\nabla f(\boldsymbol{X}_k)$ at each iteration.

---

**Algorithm 2** Full-batch Adafactor

---

**Input:** Initialization point $\boldsymbol{X}_1 \in \mathbb{R}^{n \times m}$, $\bar{\boldsymbol{R}}_0 = \boldsymbol{0}_n, \bar{\boldsymbol{C}}_0 = \boldsymbol{0}_m^\top$, step-size parameters $\{\rho_k\}_{k \geq 1}$, decay rate $\{\beta_{2,k}\}_{k \geq 1} \in [0, 1)$, regularization constant $\epsilon_1 > 0$, clipping threshold $d$.
**for** $k = 1, \cdots, T$ **do**
$\quad \bar{\boldsymbol{G}}_k = \nabla f(\boldsymbol{X}_k)$;
$\quad \bar{\boldsymbol{R}}_k = \beta_{2,k}\bar{\boldsymbol{R}}_{k-1} + (1 - \beta_{2,k})(\bar{\boldsymbol{G}}_k \odot \bar{\boldsymbol{G}}_k + \epsilon_1 \boldsymbol{1}_n \boldsymbol{1}_m^\top)\boldsymbol{1}_m$;
$\quad \bar{\boldsymbol{C}}_k = \beta_{2,k}\bar{\boldsymbol{C}}_{k-1} + (1 - \beta_{2,k})\boldsymbol{1}_n^\top(\bar{\boldsymbol{G}}_k \odot \bar{\boldsymbol{G}}_k + \epsilon_1 \boldsymbol{1}_n \boldsymbol{1}_m^\top)$;
$\quad \bar{\boldsymbol{W}}_k = (\bar{\boldsymbol{R}}_k \bar{\boldsymbol{C}}_k)/\boldsymbol{1}_n^\top \bar{\boldsymbol{R}}_k$;
$\quad \bar{\boldsymbol{U}}_k = \bar{\boldsymbol{G}}_k / \sqrt{\bar{\boldsymbol{W}}_k}$;
$\quad \hat{\eta}_k = \rho_k / \max\{1, \text{RMS}(\bar{\boldsymbol{U}}_k)/d\}$;
$\quad \boldsymbol{X}_{k+1} = \boldsymbol{X}_k - \hat{\eta}_k \cdot \bar{\boldsymbol{G}}_k / \sqrt{\bar{\boldsymbol{W}}_k}$;
**end for**

---

## A.1  Preliminary

We first denote the auxiliary matrix $\bar{\boldsymbol{G}}_{k,\epsilon_1}^2 = \bar{\boldsymbol{G}}_k \odot \bar{\boldsymbol{G}}_k + \epsilon_1 \boldsymbol{1}_n \boldsymbol{1}_m^\top$. In addition, we define $\bar{\boldsymbol{V}}_k = \left(\bar{v}_{ij}^{(k)}\right)_{ij}$ as follows,

$$\bar{\boldsymbol{V}}_0 = \boldsymbol{0}_{n \times m}, \quad \bar{\boldsymbol{V}}_k = \beta_{2,k}\bar{\boldsymbol{V}}_{k-1} + (1 - \beta_{2,k})\bar{\boldsymbol{G}}_{k,\epsilon_1}^2, \quad k \geq 1. \tag{20}$$

To simplify the notation, we let $\bar{\boldsymbol{G}}_k = \left(\bar{g}_{ij}^{(k)}\right)_{ij}$, $R_{\bar{\boldsymbol{V}}_k}^{(i)}, C_{\bar{\boldsymbol{V}}_k}^{(j)}$ and $S_{\bar{\boldsymbol{V}}_k}$ be the $i$-th row sum, $j$-th column sum and the coordinate sum of $\bar{\boldsymbol{V}}_k$ respectively. The same definition principal is applied to the notation $R_{\bar{\boldsymbol{G}}_{k,\epsilon_1}^2}^{(i)}$ and $C_{\bar{\boldsymbol{G}}_{k,\epsilon_1}^2}^{(j)}$. We also use $\bar{w}_{ij}^{(k)}, \bar{v}_{ij}^{(k)}, \bar{u}_{ij}^{(k)}$ to denote the coordinates of $\bar{\boldsymbol{W}}_k, \bar{\boldsymbol{V}}_k, \bar{\boldsymbol{U}}_k$ in Algorithm 2 respectively. In addition, we define the temporary upper bound for the gradient magnitude

$$D_t := \max_{k \in [t]} \|\bar{\boldsymbol{G}}_k\|_F, \quad \Delta_t := D_t^2 + mn\epsilon_1. \tag{21}$$

## A.2  Technical lemmas

Before proving the main result, we introduce some technical lemmas.

**Lemma A.1.** *For any $t \geq 1$, $\sum_{k=1}^t \frac{1}{k} \leq 1 + \log t$.*

*Proof.* With a simple calculation, we have

$$\sum_{k=1}^t \frac{1}{k} = 1 + \sum_{k=2}^t \int_{k-1}^k \frac{1}{k}\mathrm{d}x \leq 1 + \int_1^t \frac{1}{x}\mathrm{d}x = 1 + \log t.$$

$\square$

The following result is standard in the analysis of smooth-based optimization.

**Lemma A.2.** *Let $f$ satisfy Assumptions (A1) and (A2). Then, $\|\nabla f(\boldsymbol{X})\|_F^2 \leq 2L(f(\boldsymbol{X}) - f^*)$ and*

$$f(\boldsymbol{Y}) \leq f(\boldsymbol{X}) + \langle \nabla f(\boldsymbol{X}), \boldsymbol{Y} - \boldsymbol{X} \rangle + \frac{L}{2}\|\boldsymbol{Y} - \boldsymbol{X}\|_F^2, \quad \forall \boldsymbol{X}, \boldsymbol{Y} \in \mathbb{R}^{n \times m}. \tag{22}$$

**Lemma A.3.** *Let $\beta_{2,k} \in [0, 1], \forall k \geq 1$ and $\Gamma_k$ be defined by*

$$\Gamma_0 = 0, \quad \Gamma_k = \beta_{2,k}\Gamma_{k-1} + (1 - \beta_{2,k}), \quad \forall k \geq 1.$$

*Then, $(1 - \beta_{2,1}) \leq \Gamma_k \leq 1, \forall k \geq 1$.*

*Proof.* We could prove the result by induction. Since $\Gamma_0 = 0$, it's easy to derive that $(1 - \beta_{2,1}) = \Gamma_1 \leq 1$. Suppose that for any $j \in [k-1]$, $(1 - \beta_{2,1}) \leq \Gamma_j \leq 1$. Then

$$\Gamma_k \geq \beta_{2,k}(1 - \beta_{2,1}) + (1 - \beta_{2,k}) \geq 1 - \beta_{2,1}, \quad \Gamma_k \leq \beta_{2,k} + (1 - \beta_{2,k}) = 1.$$

The induction is then complete. $\qquad\square$

**Lemma A.4.** *Let $\bar{V}_k$ be defined in (20), $\bar{R}_k$ and $\bar{C}_k$ be defind in Algorithm 2. For any $k \geq 0$, it holds that*

$$\bar{R}_k = \bar{V}_k \mathbf{1}_m, \quad \bar{C}_k = \mathbf{1}_n^\top \bar{V}_k, \quad S_{\bar{V}_k} = \mathbf{1}_n^\top \bar{R}_k = \mathbf{1}_n^\top \bar{V}_k \mathbf{1}_m.$$

*As a consequence, for any $i \in [n], j \in [m]$,*

$$R_{\bar{V}_k}^{(i)} = \beta_{2,k} R_{\bar{V}_{k-1}}^{(i)} + (1 - \beta_{2,k}) R_{\bar{G}_{k,\epsilon_1}^2}^{(i)}, \quad C_{\bar{V}_k}^{(j)} = \beta_{2,k} C_{\bar{V}_{k-1}}^{(j)} + (1 - \beta_{2,k}) C_{\bar{G}_{k,\epsilon_1}^2}^{(j)}.$$

*Proof.* Note that $\bar{R}_0 = \bar{V}_0 \mathbf{1}_m = \mathbf{0}_n$ and $\bar{C}_0 = \mathbf{1}_n^\top \bar{V}_0 = \mathbf{0}_m^\top$. Suppose that for any $j \leq k - 1$, $\bar{R}_j = \bar{V}_j \mathbf{1}_m, \bar{C}_j = \mathbf{1}_n^\top \bar{V}_j$. Then, using the updated rule in Algorithm 2 and (20),

$$\bar{R}_k = \beta_{2,k} \bar{R}_{k-1} + (1 - \beta_{2,k}) \bar{G}_{k,\epsilon_1}^2 \mathbf{1}_m = \left(\beta_{2,k} \bar{V}_{k-1} + (1 - \beta_{2,k}) \bar{G}_{k,\epsilon_1}^2\right) \mathbf{1}_m = \bar{V}_k \mathbf{1}_m,$$

$$\bar{C}_k = \beta_{2,k} \bar{C}_{k-1} + (1 - \beta_{2,k}) \mathbf{1}_n^\top \bar{G}_{k,\epsilon_1}^2 = \mathbf{1}_n^\top \left(\beta_{2,k} \bar{V}_{k-1} + (1 - \beta_{2,k}) \bar{G}_{k,\epsilon_1}^2\right) = \mathbf{1}_n^\top \bar{V}_k. \tag{23}$$

Since $S_{\bar{V}_k}$ represents the coordinate sum of $\bar{V}_k$, we could derive that

$$S_{\bar{V}_k} = \sum_{i=1}^n \sum_{j=1}^m \bar{v}_{ij}^{(k)} = \mathbf{1}_n^\top \bar{R}_k = \mathbf{1}_n^\top \bar{V}_k \mathbf{1}_m.$$

Since $R_{\bar{V}_k}^{(i)}$ denotes the $i$-th row sum of $\bar{V}_k$, it's the $i$-th coordinate of $\bar{R}_k$. Hence, for each coordinate of $\bar{R}_k$, using (23), we get that

$$R_{\bar{V}_k}^{(i)} = \beta_{2,k} R_{\bar{V}_{k-1}}^{(i)} + (1 - \beta_{2,k}) R_{\bar{G}_{k,\epsilon_1}^2}^{(i)}.$$

Similarly, we can derive the result related to $C_{\bar{V}_k}^{(j)}$. $\qquad\square$

**Lemma A.5.** *Let $D_k$ and $\Delta_k$ be defined in (21). Then, for any $i \in [n], j \in [m], k \geq 1$, it holds that*

$$R_{\bar{V}_k}^{(i)} \in [m\epsilon_1(1 - \beta_{2,1}), D_k^2 + m\epsilon_1], \quad C_{\bar{V}_k}^{(j)} \in [n\epsilon_1(1 - \beta_{2,1}), D_k^2 + n\epsilon_1],$$

$$S_{\bar{V}_k} \in [mn\epsilon_1(1 - \beta_{2,1}), \Delta_k].$$

*Proof.* Recalling the definition of $\bar{V}_k$ in (20) and $\Gamma_k$ in Lemma A.3, we derive that

$$S_{\bar{V}_k} = \sum_{i=1}^n \sum_{j=1}^m \bar{v}_{ij}^{(k)} = \sum_{i=1}^n \sum_{j=1}^m \sum_{p=1}^k (1 - \beta_{2,p}) \left(\left(\bar{g}_{ij}^{(p)}\right)^2 + \epsilon_1\right) \left(\prod_{l=p+1}^k \beta_{2,l}\right)$$

$$\leq \sum_{p=1}^k (1 - \beta_{2,p}) \left(\prod_{l=p+1}^k \beta_{2,l}\right) \|\bar{G}_p\|_F^2 + \Gamma_k mn\epsilon_1$$

$$\leq \Gamma_k (D_k^2 + mn\epsilon_1) \leq \Delta_k, \tag{24}$$

where the last inequality applies Lemma A.3. Following (24) and Lemma A.3, we also derive that

$$S_{\bar{V}_k} \geq mn\epsilon_1 \Gamma_k \geq mn\epsilon_1(1 - \beta_{2,1}).$$

We also derive the upper bounds for $R_{\bar{V}_k}^{(i)}$ and $C_{\bar{V}_k}^{(j)}$ as follows,

$$R_{\bar{V}_k}^{(i)} = \sum_{j=1}^m \bar{v}_{ij}^{(k)} \leq \sum_{p=1}^k (1 - \beta_{2,p}) \left(\prod_{l=p+1}^k \beta_{2,l}\right) \|\bar{G}_p\|_F^2 + \Gamma_k m\epsilon_1 \leq D_k^2 + m\epsilon_1,$$

$$C_{\bar{V}_k}^{(j)} = \sum_{i=1}^n \bar{v}_{ij}^{(k)} \leq \sum_{p=1}^k (1 - \beta_{2,p}) \left(\prod_{l=p+1}^k \beta_{2,l}\right) \|\bar{G}_p\|_F^2 + \Gamma_k n\epsilon_1 \leq D_k^2 + n\epsilon_1.$$

Similarly, the lower bound could be derived by

$$R_{\bar{V}_k}^{(i)} \geq m\epsilon_1 \Gamma_k \geq m\epsilon_1(1 - \beta_{2,1}), \quad C_{\bar{V}_k}^{(j)} \geq n\epsilon_1 \Gamma_k \geq n\epsilon_1(1 - \beta_{2,1}).$$

$\qquad\square$

### A.3 Non-increasing function value.

Before proving Theorem 5.1, we need to establish a key proposition as follows, indicating that the objective function value is non-increasing under the proper selection of $\epsilon_1$ and $\rho_k$ in (4). The proof will rely on an induction argument.

**Proposition A.1.** *Following the same conditions in Theorem 5.1, for any $k \geq 1$,*

$$f(\boldsymbol{X}_{k+1}) \leq f(\boldsymbol{X}_k) - \frac{\rho_k \|\bar{\boldsymbol{G}}_k\|_F^2}{2G\Delta}, \tag{25}$$

*where $G$ and $\Delta$ are as in* (3).

*Proof.* Using Lemma A.2 and the updated rule in Algorithm 2, we get that

$$f(\boldsymbol{X}_{k+1}) \leq f(\boldsymbol{X}_k) + \langle \bar{\boldsymbol{G}}_k, \boldsymbol{X}_{k+1} - \boldsymbol{X}_k \rangle + \frac{L}{2}\|\boldsymbol{X}_{k+1} - \boldsymbol{X}_k\|_F^2$$

$$= f(\boldsymbol{X}_k) - \hat{\eta}_k \left\langle \bar{\boldsymbol{G}}_k, \frac{\bar{\boldsymbol{G}}_k}{\sqrt{\bar{\boldsymbol{W}}_k}} \right\rangle + \frac{L\hat{\eta}_k^2}{2} \left\| \frac{\bar{\boldsymbol{G}}_k}{\sqrt{\bar{\boldsymbol{W}}_k}} \right\|_F^2$$

$$\leq f(\boldsymbol{X}_k) - \underbrace{\hat{\eta}_k \left\| \frac{\bar{\boldsymbol{G}}_k}{\sqrt[4]{\bar{\boldsymbol{W}}_k}} \right\|_F^2}_{(\mathbf{a})} + \underbrace{\frac{L}{2}\hat{\eta}_k^2 \left\| \frac{\bar{\boldsymbol{G}}_k}{\sqrt{\bar{\boldsymbol{W}}_k}} \right\|_F^2}_{(\mathbf{b})}. \tag{26}$$

**Step 1: Estimating (a) and (b).** To lower bound **(a)**, we first discuss the maximum operator inside $\hat{\eta}_k$. Let two index sets be defined as

$$E_1^{(k)} = \left\{ s \in [k] \mid \|\bar{\boldsymbol{U}}_s\|_F \geq d\sqrt{mn} \right\}, \quad E_2^{(k)} = \left\{ s \in [k] \mid \|\bar{\boldsymbol{U}}_s\|_F < d\sqrt{mn} \right\}.$$

Using Lemma A.5 and $w_{ij}^{(k)} = \frac{R_{\bar{\boldsymbol{V}}_k}^{(i)} C_{\bar{\boldsymbol{V}}_k}^{(j)}}{S_{\bar{\boldsymbol{V}}_k}}$, and noting that $R_{\bar{\boldsymbol{V}}_k}^{(i)}, C_{\bar{\boldsymbol{V}}_k}^{(j)} \leq S_{\bar{\boldsymbol{V}}_k}$, we derive that

$$\bar{w}_{ij}^{(k)} \geq \frac{mn\epsilon_1^2(1-\beta_{2,1})^2}{\Delta_k}, \quad w_{ij}^{(k)} \leq S_{\bar{\boldsymbol{V}}_k} \leq \Delta_k. \tag{27}$$

Then, we have

$$\|\bar{\boldsymbol{U}}_k\|_F^2 = \sum_{i=1}^n \sum_{j=1}^m \frac{\left(\bar{g}_{ij}^{(k)}\right)^2}{\bar{w}_{ij}^{(k)}} \leq \frac{\|\bar{\boldsymbol{G}}_k\|_F^2 \Delta_k}{mn\epsilon_1^2(1-\beta_{2,1})^2} \leq \frac{D_k^2 \Delta_k}{mn\epsilon_1^2(1-\beta_{2,1})^2}. \tag{28}$$

Hence, when $k \in E_1^{(t)}$, the clipping is effective and we get that

$$\hat{\eta}_k \left\| \frac{\bar{\boldsymbol{G}}_k}{\sqrt[4]{\bar{\boldsymbol{W}}_k}} \right\|_F^2 \geq \frac{d\sqrt{mn}\rho_k}{\|\bar{\boldsymbol{U}}_k\|_F} \frac{\|\bar{\boldsymbol{G}}_k\|_F^2}{\max_{i,j}\sqrt{\bar{w}_{ij}^{(k)}}} \geq d\epsilon_1 mn(1-\beta_{2,1})\frac{\rho_k \|\bar{\boldsymbol{G}}_k\|_F^2}{D_k \Delta_k}. \tag{29}$$

When $k \in E_2^{(t)}$, the clipping does not work and we obtain that

$$\hat{\eta}_k \left\| \frac{\bar{\boldsymbol{G}}_k}{\sqrt[4]{\bar{\boldsymbol{W}}_k}} \right\|_F^2 \geq \frac{\rho_k \|\bar{\boldsymbol{G}}_k\|_F^2}{\max_{i,j}\sqrt{\bar{w}_{ij}^{(k)}}} \geq \frac{\rho_k \|\bar{\boldsymbol{G}}_k\|_F^2}{\sqrt{\Delta_k}}. \tag{30}$$

Combining with (29) and (30), and using $\epsilon_1 = \frac{c_0}{dmn(1-\beta_{2,1})}$, we derive that

$$(\mathbf{a}) \geq \min\left\{ \frac{1}{\sqrt{\Delta_k}}, \frac{c_0}{D_k \Delta_k} \right\} \rho_k \|\bar{\boldsymbol{G}}_k\|_F^2. \tag{31}$$

Using (27), we have

$$(\mathbf{b}) \leq \frac{L\rho_k^2 \|\bar{\boldsymbol{G}}_k\|_F^2}{2\min_{i,j}\bar{w}_{ij}^{(k)}} \leq \frac{L\rho_k^2 \|\bar{\boldsymbol{G}}_k\|_F^2 \Delta_k}{2(1-\beta_{2,1})^2 mn\epsilon_1^2}. \tag{32}$$

**Step 2: Verifying $k = 1$.** To prove the desired result in (25), we use an induction argument. First, we need to prove the case of $k = 1$. Note that when $k = 1$, from Lemma A.2, $\epsilon_1$ in (4) and $D_1, \Delta_1$ defined in (21), we get that

$$D_1^2 = \|\bar{\boldsymbol{G}}_1\|_F^2 \leq 2L(f(\boldsymbol{X}_1) - f^*) \leq G^2, \quad \Delta_1 = D_1^2 + mn\epsilon_1 \leq G^2 + mn\epsilon_1 \leq \Delta. \tag{33}$$

Then, setting $k = 1$ in (31) and using (33),

$$(\mathbf{a}) \geq \min\left\{\frac{1}{\sqrt{\Delta}}, \frac{c_0}{G\Delta}\right\} \rho_1 \|\bar{\boldsymbol{G}}_1\|_F^2 = \frac{c_0\rho_1\|\bar{\boldsymbol{G}}_1\|_F^2}{G\Delta}, \tag{34}$$

where the equality applies that $\Delta \geq 1$ and $\frac{c_0}{G} \leq 1$ from (3). Similarly, applying (33) into (32) with $k = 1$, and combining with (34) and (26) with $k = 1$,

$$f(\boldsymbol{X}_2) \leq f(\boldsymbol{X}_1) + \rho_1\|\bar{\boldsymbol{G}}_1\|_F^2 \left(\frac{L\rho_1\Delta}{2(1-\beta_{2,1})^2 mn\epsilon_1^2} - \frac{c_0}{G\Delta}\right) \leq f(\boldsymbol{X}_1) - \frac{c_0\rho_1\|\bar{\boldsymbol{G}}_1\|_F^2}{2G\Delta},$$

where the last inequality applies the setups of $\epsilon_1, \rho_1$ in (4).

**Step 3: Verifying $k = t$.** Suppose that for any $k \leq t - 1$, (25) holds. Consequently, for any $k \leq t$,

$$\|\bar{\boldsymbol{G}}_k\|_F^2 \leq 2L(f(\boldsymbol{X}_k) - f^*) \leq \cdots \leq 2L(f(\boldsymbol{X}_1) - f^*) \leq G^2, \quad \Delta_k \leq \Delta. \tag{35}$$

Then, setting $k = t$ in (31) and (32), and using (35), we have

$$(\mathbf{a}) \geq \min\left\{\frac{1}{\sqrt{\Delta}}, \frac{c_0}{G\Delta}\right\} \rho_t \|\bar{\boldsymbol{G}}_t\|_F^2 = \frac{c_0\rho_t\|\bar{\boldsymbol{G}}_t\|_F^2}{G\Delta}, \quad (\mathbf{b}) \leq \frac{L\rho_t^2\|\bar{\boldsymbol{G}}_t\|_F^2\Delta}{2(1-\beta_{2,1})^2 mn\epsilon_1^2}. \tag{36}$$

Plugging (36) into (26) with $k = t$, and using $\rho_t = \rho_0$ in (4), we get that

$$f(\boldsymbol{X}_{t+1}) \leq f(\boldsymbol{X}_t) + \rho_t\|\bar{\boldsymbol{G}}_t\|_F^2 \left(\frac{L\rho_t\Delta}{2(1-\beta_{2,1})^2 mn\epsilon_1^2} - \frac{c_0}{G\Delta}\right) \leq f(\boldsymbol{X}_t) - \frac{c_0\rho_t\|\bar{\boldsymbol{G}}_t\|_F^2}{2G\Delta}.$$

Then, the induction is complete, and we prove the desired result. $\qquad\square$

### A.4   Proof of Theorem 5.1

Now, based on Proposition A.1, we can easily prove the main convergence result. Consequently, subtracting $f^*$ on both sides of (25) and summing up both sides over $k \in [T]$,

$$\sum_{k=1}^{T} \frac{\rho_k\|\bar{\boldsymbol{G}}_k\|_F^2}{2G\Delta} \leq f(\boldsymbol{X}_1) - f(\boldsymbol{X}_{t+1}) \leq f(\boldsymbol{X}_1) - f^*,$$

where the last inequality applies Assumption (A2). Then, with $\rho_k = \rho_0$, we can derive that

$$\min_{k \in [T]} \|\bar{\boldsymbol{G}}_k\|_F^2 \leq \frac{1}{T}\sum_{k=1}^{T} \|\bar{\boldsymbol{G}}_k\|_F^2 \leq \frac{2G\Delta\left(f(\boldsymbol{X}_1) - f^*\right)}{\rho_0 T}.$$

## B   Proof detail for Theorem 6.1

### B.1   Preliminary

We first follow the notations of $\bar{\boldsymbol{G}}_k = \left(\bar{g}_{ij}^{(k)}\right)_{ij} := \nabla f(\boldsymbol{X}_k)$. Let $\boldsymbol{G}_k = \left(g_{ij}^{(k)}\right)_{ij}$ and $\boldsymbol{\xi}_k := \boldsymbol{G}_k - \bar{\boldsymbol{G}}_k$. We define $\boldsymbol{G}_{k,\epsilon_1}^2 := \boldsymbol{G}_k \odot \boldsymbol{G}_k + \epsilon_1 \mathbf{1}_n \mathbf{1}_m^\top$ and $\boldsymbol{V}_k = \left(v_{ij}^{(k)}\right)_{ij}$ such that

$$\boldsymbol{V}_0 = \mathbf{0}_{n \times m}, \quad \boldsymbol{V}_k = \beta_{2,k}\boldsymbol{V}_{k-1} + (1 - \beta_{2,k})\boldsymbol{G}_{k,\epsilon_1}^2, \quad k \geq 1. \tag{37}$$

We also define $R_{\boldsymbol{V}_k}^{(i)}, C_{\boldsymbol{V}_k}^{(j)}$ and $S_{\boldsymbol{V}_k}$ as the $i$-th row sum, $j$-th column sum and coordinate sum of $\boldsymbol{V}_k$ respectively. $R_{\boldsymbol{G}_{k,\epsilon_1}^2}^{(i)}$ and $C_{\boldsymbol{G}_{k,\epsilon_1}^2}^{(j)}$ represent the same definitions with respect to $\boldsymbol{G}_{k,\epsilon_1}^2$. Then, using a similar deduction in Lemma A.4, we obtain that for any $k \geq 1, i \in [n], j \in [m]$,

$$R_{\boldsymbol{V}_k}^{(i)} = \beta_{2,k}R_{\boldsymbol{V}_{k-1}}^{(i)} + (1 - \beta_{2,k})R_{\boldsymbol{G}_{k,\epsilon_1}^2}^{(i)}, \quad C_{\boldsymbol{V}_k}^{(j)} = \beta_{2,k}C_{\boldsymbol{V}_{k-1}}^{(j)} + (1 - \beta_{2,k})C_{\boldsymbol{G}_{k,\epsilon_1}^2}^{(j)}. \tag{38}$$

As a consequence of (38), each coordinate of $\boldsymbol{W}_k$ satisfies that

$$w_{ij}^{(k)} = \frac{R_{\boldsymbol{V}_k}^{(i)} C_{\boldsymbol{V}_k}^{(j)}}{S_{\boldsymbol{V}_k}} = \frac{\left(\beta_{2,k} R_{\boldsymbol{V}_{k-1}}^{(i)} + (1-\beta_{2,k}) R_{\boldsymbol{G}_{k,\epsilon_1}^2}^{(i)}\right)\left(\beta_{2,k} C_{\boldsymbol{V}_{k-1}}^{(j)} + (1-\beta_{2,k}) C_{\boldsymbol{G}_{k,\epsilon_1}^2}^{(j)}\right)}{\beta_{2,k} S_{\boldsymbol{V}_{k-1}} + (1-\beta_{2,k}) S_{\boldsymbol{G}_{k,\epsilon_1}^2}}.$$

(39)

**A well-constructed proxy step-size.** For any $k \geq 1$, define

$$D_k := \max_{s \in [k]} \|\bar{\boldsymbol{G}}_s\|_F, \quad \Sigma_k := D_k + \sigma\sqrt{\log\left(\frac{\mathrm{e}T}{\delta}\right)},$$

$$\mathcal{G}_{k,1} := \Sigma_k^2 + m\epsilon_1, \quad \mathcal{G}_{k,2} := \Sigma_k^2 + n\epsilon_1, \quad \mathcal{G}_k := \Sigma_k^2 + mn\epsilon_1.$$

(40)

Then, we introduce a proxy step-size matrix $\boldsymbol{A}_k = \left(a_{ij}^{(k)}\right)_{ij}$ such that

$$a_{ij}^{(k)} = \frac{\left(\beta_{2,k} R_{\boldsymbol{V}_{k-1}}^{(i)} + (1-\beta_{2,k})\mathcal{G}_{k,1}\right)\left(\beta_{2,k} C_{\boldsymbol{V}_{k-1}}^{(j)} + (1-\beta_{2,k})\mathcal{G}_{k,2}\right)}{\beta_{2,k} S_{\boldsymbol{V}_{k-1}} + (1-\beta_{2,k})\mathcal{G}_k}.$$

(41)

The proxy step-size technique is a standard way in the convergence analysis of adaptive methods, e.g., [44, 11]. We provide a new proxy step-size in (41) to handle the matrix factorization in Adafactor. This construction satisfies two properties. First, it's independent from the $k$-th random sample $\boldsymbol{Z}_k$ and thereby conditionally independent with the $k$-th stochastic gradient $\boldsymbol{G}_k$. Second, it needs to remain sufficiently close to the original adaptive step-size $\boldsymbol{W}_k$ to avoid generating divergent terms, as indicated in Lemma B.6.

## B.2 Technical lemmas

In the following, we first provide some necessary technical lemmas. We introduce a concentration inequality for the martingale difference sequence. See [27] for a proof.

**Lemma B.1.** *Suppose that $\{Z_s\}_{s \in [T]}$ is a martingale difference sequence with respect to $\zeta_1, \cdots, \zeta_T$. Assume that for each $s \in [T]$, $\sigma_s$ is a random variable only dependent on $\zeta_1, \cdots, \zeta_{s-1}$ and satisfies that*

$$\mathbb{E}\left[\exp\left(\frac{Z_s^2}{\sigma_s^2}\right)\Big|\zeta_1, \cdots, \zeta_{s-1}\right] \leq \mathrm{e}.$$

*Then, for any $\lambda > 0$, and for any $\delta \in (0,1)$, it holds that*

$$\mathbb{P}\left(\sum_{s=1}^{T} Z_s > \frac{1}{\lambda}\log\left(\frac{1}{\delta}\right) + \frac{3}{4}\lambda \sum_{s=1}^{T}\sigma_s^2\right) \leq \delta.$$

We also introduce a standard result showing that the maximum magnitude of a sequence of vectors with sub-Gaussian norm is restricted. See [27, Lemma 5] for a proof.

**Lemma B.2.** *Let $T \geq 1$ and $\boldsymbol{\xi}_k = \boldsymbol{G}_k - \bar{\boldsymbol{G}}_k, \forall k \in [T]$ satisfy Assumption (A4). Then, with probability at least $1 - \delta$,*

$$\max_{k \in [T]} \|\boldsymbol{\xi}_k\|_F^2 \leq \sigma^2 \log\left(\frac{\mathrm{e}T}{\delta}\right).$$

(42)

Then, the following lemmas will be established based on the probabilistic event in Lemma B.2.

**Lemma B.3.** *Let $T \geq 1, \beta_{2,1} = 1/2, \beta_{2,k} \in [0,1), \forall k \geq 2$ and $\mathcal{G}_{k,1}, \mathcal{G}_{k,2}, \mathcal{G}_k$ be defined in (40). If (42) happens, then, for any $k \in [T]$, $i \in [n]$ and $j \in [m]$,*

$$R_{\boldsymbol{G}_{k,\epsilon_1}^2}^{(i)}, R_{\boldsymbol{V}_k}^{(i)} \in [m\epsilon_1/2, \mathcal{G}_{k,1}], \quad C_{\boldsymbol{G}_{k,\epsilon_1}^2}^{(j)}, C_{\boldsymbol{V}_k}^{(j)} \in [n\epsilon_1/2, \mathcal{G}_{k,2}], \quad S_{\boldsymbol{G}_{k,\epsilon_1}^2}, S_{\boldsymbol{V}_k} \in [mn\epsilon_1/2, \mathcal{G}_k].$$

*Proof.* First, using (42), we have for any $k \in [T]$,

$$\|\boldsymbol{G}_k\|_F \leq \|\bar{\boldsymbol{G}}_k\|_F + \|\boldsymbol{\xi}_k\|_F \leq D_k + \sigma \sqrt{\log\left(\frac{eT}{\delta}\right)} = \Sigma_k. \tag{43}$$

Using (40), we derive that

$$mn\epsilon_1/2 \leq S_{\boldsymbol{G}^2_{k,\epsilon_1}} = \sum_{i=1}^{n}\sum_{j=1}^{m}\left(\left(g_{ij}^{(k)}\right)^2 + \epsilon_1\right) = \|\boldsymbol{G}_k\|_F^2 + mn\epsilon_1 \leq \mathcal{G}_k,$$

$$m\epsilon_1/2 \leq R_{\boldsymbol{G}^2_{k,\epsilon_1}}^{(i)} = \sum_{j=1}^{m}\left(\left(g_{ij}^{(k)}\right)^2 + \epsilon_1\right) \leq \|\boldsymbol{G}_k\|_F^2 + m\epsilon_1 \leq \mathcal{G}_{k,1},$$

$$n\epsilon_1/2 \leq C_{\boldsymbol{G}^2_{k,\epsilon_1}}^{(j)} = \sum_{i=1}^{n}\left(\left(g_{ij}^{(k)}\right)^2 + \epsilon_1\right) \leq \|\boldsymbol{G}_k\|_F^2 + n\epsilon_1 \leq \mathcal{G}_{k,2}.$$

Using Lemma A.3 and (43), we can show that

$$m\epsilon_1(1 - \beta_{2,1}) \leq R_{\boldsymbol{V}_k}^{(i)} \leq \sum_{p=1}^{k}(1 - \beta_{2,p})\left(\prod_{l=p+1}^{k}\beta_{2,l}\right)\|\boldsymbol{G}_p\|_F^2 + \Gamma_k m\epsilon_1 \leq \Gamma_k(\Sigma_k^2 + m\epsilon_1).$$

With $\beta_{2,1} = 1/2$, we then obtain the desired result. The bounds for $C_{\boldsymbol{V}_k}^{(j)}, S_{\boldsymbol{V}_k}$ can be also derived by the similar deduction. $\square$

We have the following lemma to control each coordinate of the proxy step-size matrix $\boldsymbol{A}_k$.

**Lemma B.4.** *Let* $T \geq 1, \beta_{2,1} = 1/2, \beta_{2,k} \in [0, 1), \forall k \geq 2$. *If* (42) *happens, then it holds that for any* $k \in [T], i \in [n], j \in [m]$,

$$\frac{mn\epsilon_1^2}{4\mathcal{G}_k} \leq w_{ij}^{(k)}, \quad \frac{mn\epsilon_1^2}{4\mathcal{G}_k} \leq a_{ij}^{(k)} \leq \min\{\mathcal{G}_{k,1}, \mathcal{G}_{k,2}\}.$$

*Consequently,* $\left\|\frac{\boldsymbol{G}_k}{\sqrt{\boldsymbol{W}_k}}\right\|_F^2 \leq \frac{4\Sigma_k^2 \mathcal{G}_k}{mn\epsilon_1^2}.$

*Proof.* With $w_{ij}^{(k)} = \frac{R_{\boldsymbol{V}_k}^{(i)}C_{\boldsymbol{V}_k}^{(j)}}{S_{\boldsymbol{V}_k}}$, we can easily derive from Lemma B.3 that

$$w_{ij}^{(k)} \geq \frac{mn\epsilon_1^2}{4\mathcal{G}_k}, \quad \left\|\frac{\boldsymbol{G}_k}{\sqrt{\boldsymbol{W}_k}}\right\|_F^2 \leq \frac{\|\boldsymbol{G}_k\|_F^2}{\min_{i,j} w_{ij}^{(k)}} \leq \frac{4\Sigma_k^2 \mathcal{G}_k}{mn\epsilon_1^2},$$

where the last inequality applies (43). Since $R_{\boldsymbol{V}_{k-1}}^{(i)}, C_{\boldsymbol{V}_{k-1}}^{(j)} \leq S_{\boldsymbol{V}_{k-1}}$ and $\mathcal{G}_{k,1}, \mathcal{G}_{k,2} \leq \mathcal{G}_k$, we have

$$\frac{\beta_{2,k}R_{\boldsymbol{V}_{k-1}}^{(i)} + (1 - \beta_{2,k})\mathcal{G}_{k,1}}{\beta_{2,k}S_{\boldsymbol{V}_{k-1}} + (1 - \beta_{2,k})\mathcal{G}_k} \leq 1, \quad \frac{\beta_{2,k}C_{\boldsymbol{V}_{k-1}}^{(j)} + (1 - \beta_{2,k})\mathcal{G}_{k,2}}{\beta_{2,k}S_{\boldsymbol{V}_{k-1}} + (1 - \beta_{2,k})\mathcal{G}_k} \leq 1.$$

Then, using Lemma B.3, we derive that

$$a_{ij}^{(k)} \leq \min\left\{\beta_{2,k}R_{\boldsymbol{V}_{k-1}}^{(i)} + (1 - \beta_{2,k})\mathcal{G}_{k,1}, \beta_{2,k}C_{\boldsymbol{V}_{k-1}}^{(j)} + (1 - \beta_{2,k})\mathcal{G}_{k,2}\right\} \leq \min\{\mathcal{G}_{k,1}, \mathcal{G}_{k,2}\}. \tag{44}$$

To lower bound $a_{ij}^{(k)}$, we can derive from Lemma B.3 that

$$\beta_{2,k}R_{\boldsymbol{V}_{k-1}}^{(i)} + (1 - \beta_{2,k})\mathcal{G}_{k,1} \geq \beta_{2,k}(m\epsilon_1/2) + (1 - \beta_{2,k})(m\epsilon_1/2) = m\epsilon_1/2.$$

Similarly, we get that $\beta_{2,k}C_{\boldsymbol{V}_{k-1}}^{(j)} + (1 - \beta_{2,k})\mathcal{G}_{k,2} \geq n\epsilon_1/2$ and further deriv that $a_{ij}^{(k)} \geq \frac{m\epsilon_1 \cdot n\epsilon_1}{4\mathcal{G}_k}$. $\square$

Next, we have the following probabilistic result relying on the property of the martingale difference sequence and sub-Gaussian noise.

**Lemma B.5.** *Let $\rho_k$ be defined in (7) and $\beta_{2,k} \in [0,1)$. Let Assumptions (A3), (A4) hold and $\mathcal{H}$ be as in (5). For any $T \geq 1, \lambda > 0$ and $\delta \in (0,1)$, it holds that with probability at least $1 - \delta$,*

$$-\sum_{k=1}^{t} \rho_k \left\langle \bar{G}_k, \frac{\xi_k}{\sqrt{A_k}} \right\rangle \leq \frac{3\lambda\sigma^2}{4} \sum_{k=1}^{t} \rho_k^2 \left\| \frac{\bar{G}_k}{\sqrt{A_k}} \right\|_F^2 + \frac{1}{\lambda} \log\left(\frac{1}{\delta}\right), \quad \forall t \in [T].$$

*Proof.* Let $\zeta_k = -\rho_k \left\langle \bar{G}_k, \frac{\xi_k}{\sqrt{A_k}} \right\rangle$ and the filtration $\mathcal{F}_k = \sigma\left(Z_1, \cdots, Z_k\right)$ where $\sigma(\cdot)$ denotes the $\sigma$-algebra. Note that $\rho_k, \bar{G}_k$ and $A_k$ are measurable with $\mathcal{F}_{k-1}$ and $\xi_k$ is measurable with $\mathcal{F}_k$. Then, $\{\zeta_k\}_{k\geq 1}$ is a martingale difference sequence with $\mathcal{F}_k$ since from Assumption (A3),

$$\mathbb{E}\left[\zeta_k \mid \mathcal{F}_{k-1}\right] = -\rho_k \left\langle \bar{G}_k, \frac{\mathbb{E}\left[\xi_k \mid \mathcal{F}_{k-1}\right]}{\sqrt{A_k}} \right\rangle = 0.$$

Let $\omega_k = \sigma\rho_k \left\| \frac{\bar{G}_k}{\sqrt{A_k}} \right\|_F$. We derive from Cauchy-Schwarz inequality and Assumption (A4) that

$$\mathbb{E}\left[\exp\left(\frac{\zeta_k^2}{\omega_k^2}\right) \mid \mathcal{F}_{k-1}\right] \leq \mathbb{E}\left[\exp\left(\frac{\left\| \frac{\bar{G}_k}{\sqrt{A_k}} \right\|_F^2 \|\xi_k\|_F^2}{\sigma^2 \left\| \frac{\bar{G}_k}{\sqrt{A_k}} \right\|_F^2}\right) \Big| \mathcal{F}_{k-1}\right] \leq e. \tag{45}$$

Then, using Lemma B.1, it leads to that for any $\lambda > 0$, with probability at least $1 - \delta$,

$$-\sum_{k=1}^{t} \rho_k \left\langle \bar{G}_k, \frac{\xi_k}{\sqrt{A_k}} \right\rangle \leq \frac{3\lambda\sigma^2}{4} \sum_{k=1}^{t} \rho_k^2 \left\| \frac{\bar{G}_k}{\sqrt{A_k}} \right\|_F^2 + \frac{1}{\lambda} \log\left(\frac{1}{\delta}\right). \tag{46}$$

$\square$

The following key lemma provides an upper bound for the "relative distance" between $W_k$ and $A_k$.

**Lemma B.6.** *Let $T \geq 1, \beta_{2,1} = 1/2, \beta_{2,k} \in [0,1), \forall k \geq 2$. If (42) happens, then for any $k \geq 1, i \in [n], j \in [m]$ and $\mathcal{G}_k$ in (40), it holds that*

$$\frac{\left| w_{ij}^{(k)} - a_{ij}^{(k)} \right|}{\sqrt{a_{ij}^{(k)}}} \leq 3\sqrt{(1 - \beta_{2,k})\mathcal{G}_k}. \tag{47}$$

*Proof.* To simplify the notation, we let

$$X_k = \beta_{2,k} R_{V_{k-1}}^{(i)} + (1 - \beta_{2,k}) R_{G_{k,\epsilon_1}^2}^{(i)}, \quad \bar{X}_k = (1 - \beta_{2,k})\left(\mathcal{G}_{k,1} - R_{G_{k,\epsilon_1}^2}^{(i)}\right),$$

$$Y_k = \beta_{2,k} C_{V_{k-1}}^{(j)} + (1 - \beta_{2,k}) C_{G_{k,\epsilon_1}^2}^{(j)}, \quad \bar{Y}_k = (1 - \beta_{2,k})\left(\mathcal{G}_{k,2} - C_{G_{k,\epsilon_1}^2}^{(j)}\right),$$

$$Z_k = \beta_{2,k} S_{V_{k-1}} + (1 - \beta_{2,k}) S_{G_{k,\epsilon_1}^2}, \quad \bar{Z}_k = (1 - \beta_{2,k})\left(\mathcal{G}_k - S_{G_{k,\epsilon_1}^2}\right). \tag{48}$$

Then, we have

$$\left| w_{ij}^{(k)} - a_{ij}^{(k)} \right| = \left| \frac{X_k Y_k}{Z_k} - \frac{(X_k + \bar{X}_k)(Y_k + \bar{Y}_k)}{Z_k + \bar{Z}_k} \right|$$

$$= \left| \frac{X_k Y_k \bar{Z}_k - X_k Z_k \bar{Y}_k - Y_k Z_k \bar{X}_k - Z_k \bar{X}_k \bar{Y}_k}{Z_k(Z_k + \bar{Z}_k)} \right|.$$

Recalling $a_{ij}^{(k)}$ in (41), we get that $a_{ij}^{(k)} = \frac{(X_k + \bar{X}_k)(Y_k + \bar{Y}_k)}{Z_k + \bar{Z}_k}$. Hence, we derive that

$$\frac{\left| w_{ij}^{(k)} - a_{ij}^{(k)} \right|}{\sqrt{a_{ij}^{(k)}}} = \frac{\left| X_k Y_k \bar{Z}_k - X_k Z_k \bar{Y}_k - Y_k Z_k \bar{X}_k - Z_k \bar{X}_k \bar{Y}_k \right|}{Z_k \sqrt{(X_k + \bar{X}_k)(Y_k + \bar{Y}_k)(Z_k + \bar{Z}_k)}}$$

$$\leq \underbrace{\frac{\left| X_k \bar{Y}_k + Y_k \bar{X}_k + (\bar{X}_k \bar{Y}_k) \right|}{\sqrt{(X_k + \bar{X}_k)(Y_k + \bar{Y}_k)(Z_k + \bar{Z}_k)}}}_{\text{(c)}} + \underbrace{\frac{X_k Y_k \bar{Z}_k}{Z_k \sqrt{(X_k + \bar{X}_k)(Y_k + \bar{Y}_k)(Z_k + \bar{Z}_k)}}}_{\text{(d)}}. \tag{49}$$

Since (42) happens, we can apply Lemma B.3 to verify that

$$0 \le \bar{X}_k \le (1 - \beta_{2,k})\mathcal{G}_{k,1}, \quad 0 \le \bar{Y}_k \le (1 - \beta_{2,k})\mathcal{G}_{k,2}, \quad 0 \le \bar{Z}_k \le (1 - \beta_{2,k})\mathcal{G}_k. \quad (50)$$

Since $X_k Y_k \ge 0$ and $Z_k + \bar{Z}_k > 0$, (c) can be bounded as

$$(\mathbf{c}) \le \frac{\left| X_k \bar{Y}_k + Y_k \bar{X}_k + \bar{X}_k \bar{Y}_k \right|}{\sqrt{(X_k \bar{Y}_k + Y_k \bar{X}_k + \bar{X}_k \bar{Y}_k)(Z_k + \bar{Z}_k)}} \le \sqrt{\frac{X_k \bar{Y}_k + Y_k \bar{X}_k + \bar{X}_k \bar{Y}_k}{Z_k + \bar{Z}_k}}. \quad (51)$$

Recalling the definition, we have $X_k, \bar{X}_k \le Z_k + \bar{Z}_k$ and $Y_k \le Z_k + \bar{Z}_k$. Further, applying (50), we derive that

$$\frac{X_k \bar{Y}_k}{Z_k + \bar{Z}_k} \le \bar{Y}_k \le (1 - \beta_{2,k})\mathcal{G}_{k,2}, \quad \frac{Y_k \bar{X}_k}{Z_k + \bar{Z}_k} \le \bar{X}_k \le (1 - \beta_{2,k})\mathcal{G}_{k,1},$$

$$\frac{\bar{X}_k \bar{Y}_k}{Z_k + \bar{Z}_k} \le \bar{Y}_k \le (1 - \beta_{2,k})\mathcal{G}_{k,2}.$$

We then derive from (51), $\mathcal{G}_{k,1} \le \mathcal{G}_k$ and $\mathcal{G}_{k,2} \le \mathcal{G}_k$ that

$$(\mathbf{c}) \le \sqrt{3(1 - \beta_{2,k})\mathcal{G}_k}. \quad (52)$$

Then, we move to bound (d). Recalling the definitions in (48), we have $0 \le X_k \le Z_k, 0 \le Y_k \le Z_k$. Combining (50) where $\bar{X}_k, \bar{Y}_k, \bar{Z}_k \ge 0$, we have

$$(\mathbf{d}) \le \frac{X_k Y_k \bar{Z}_k}{Z_k \sqrt{X_k Y_k \bar{Z}_k}} \le \frac{\sqrt{X_k Y_k \bar{Z}_k}}{Z_k} \le \sqrt{\bar{Z}_k} \le \sqrt{(1 - \beta_{2,k})\mathcal{G}_k}. \quad (53)$$

Applying (52) and (53) into (49), we then derive the desired result. $\qquad\square$

## B.3 Bounding gradient magnitude

In this part, we will control the gradient magnitude along the optimization trajectory. The result is summarized in the following proposition.

**Proposition B.1.** *Following the same conditions and notations in Theorem 6.1, for any $T \ge 1$ and $\delta \in (0, 1/2)$, it holds that with probability at least $1 - 2\delta$,*

$$D_t = \max_{k \in [t]} \|\bar{G}_k\|_F \le H, \quad \Sigma_t \le \Sigma_H, \quad \mathcal{G}_t \le \mathcal{H}, \quad \forall t \in [T]. \quad (54)$$

*Proof.* Using the inequality in Lemma A.2 and Algorithm 1, we have

$$f(\boldsymbol{X}_{k+1}) \le f(\boldsymbol{X}_k) + \langle \bar{\boldsymbol{G}}_k, \boldsymbol{X}_{k+1} - \boldsymbol{X}_k \rangle + \frac{L}{2} \|\boldsymbol{X}_{k+1} - \boldsymbol{X}_k\|_F^2$$

$$\le f(\boldsymbol{X}_k) - \eta_k \left\langle \bar{\boldsymbol{G}}_k, \frac{\boldsymbol{G}_k}{\sqrt{\boldsymbol{W}_k}} \right\rangle + \frac{L\eta_k^2}{2} \left\| \frac{\boldsymbol{G}_k}{\sqrt{\boldsymbol{W}_k}} \right\|_F^2.$$

Introducing the proxy step-size matrix $\boldsymbol{A}_k$ in (41) and then summing up both sides over $k \in [t]$, we derive that

$$f(\boldsymbol{X}_{t+1}) \le f(\boldsymbol{X}_1) \underbrace{- \sum_{k=1}^t \eta_k \left\langle \bar{\boldsymbol{G}}_k, \frac{\boldsymbol{G}_k}{\sqrt{\boldsymbol{A}_k}} \right\rangle}_{\mathbf{A}}$$

$$+ \underbrace{\sum_{k=1}^t \eta_k \left\langle \bar{\boldsymbol{G}}_k, \boldsymbol{G}_k \odot \left( \frac{1}{\sqrt{\boldsymbol{A}_k}} - \frac{1}{\sqrt{\boldsymbol{W}_k}} \right) \right\rangle}_{\mathbf{B}} + \underbrace{\sum_{k=1}^t \frac{L\eta_k^2}{2} \left\| \frac{\boldsymbol{G}_k}{\sqrt{\boldsymbol{W}_k}} \right\|_F^2}_{\mathbf{C}}. \quad (55)$$

First, we will assume that the probability event in (42) happens and estimate **B, C** relying on the temporary upper bounds $D_k, \Sigma_k, \mathcal{G}_k$ in (40). The estimation for **A** is given during the induction argument. Note that when the same conditions in Theorem 6.1 hold and (42) holds, Lemmas B.3, B.4, B.6 hold. To start with, using (42), we have

$$\|\boldsymbol{G}_k\|_F \le \|\bar{\boldsymbol{G}}_k\|_F + \|\boldsymbol{\xi}_k\|_F \le \Sigma_k. \quad (56)$$

**Estimating B.** **B** is essentially the error brought by the proxy step-size $\boldsymbol{A}_k$. We will first calculate the gap of $1/\sqrt{w_{ij}^{(k)}}$ and $1/\sqrt{a_{ij}^{(k)}}$ as follows,

$$\left| \frac{1}{\sqrt{w_{ij}^{(k)}}} - \frac{1}{\sqrt{a_{ij}^{(k)}}} \right| = \frac{1}{\sqrt{w_{ij}^{(k)}}\sqrt{a_{ij}^{(k)}}} \left| \sqrt{w_{ij}^{(k)}} - \sqrt{a_{ij}^{(k)}} \right| \leq \frac{1}{\sqrt{w_{ij}^{(k)}}\sqrt{a_{ij}^{(k)}}} \sqrt{\left| w_{ij}^{(k)} - a_{ij}^{(k)} \right|}. \quad (57)$$

We then apply (57) and Young's inequality,

$$\begin{aligned}
\mathbf{B} &\leq \sum_{k=1}^{t} \sum_{i=1}^{n} \sum_{j=1}^{m} \eta_k \frac{\left| \bar{g}_{ij}^{(k)} g_{ij}^{(k)} \right|}{\sqrt{w_{ij}^{(k)}}\sqrt{a_{ij}^{(k)}}} \sqrt{\left| w_{ij}^{(k)} - a_{ij}^{(k)} \right|} \\
&\leq \frac{1}{4} \sum_{k=1}^{t} \sum_{i=1}^{n} \sum_{j=1}^{m} \eta_k \cdot \frac{\left( \bar{g}_{ij}^{(k)} \right)^2}{\sqrt{a_{ij}^{(k)}}} + 4 \sum_{k=1}^{t} \sum_{i=1}^{n} \sum_{j=1}^{m} \eta_k \cdot \frac{\left| w_{ij}^{(k)} - a_{ij}^{(k)} \right|}{\sqrt{a_{ij}^{(k)}}} \cdot \left( \frac{g_{ij}^{(k)}}{\sqrt{w_{ij}^{(k)}}} \right)^2.
\end{aligned} \quad (58)$$

Thus, plugging (47) from Lemma B.6 into (58), then using Lemma B.4 and $\eta_k = \rho_k = \rho_0/k^{1-c/2}, \beta_{2,1} = 1/2, \beta_{2,k} = 1 - 1/k^c, k \geq 2$, we derive that

$$\begin{aligned}
\mathbf{B} &\leq \frac{1}{4} \sum_{k=1}^{t} \eta_k \left\| \frac{\bar{\boldsymbol{G}}_k}{\sqrt[4]{\boldsymbol{A}_k}} \right\|_F^2 + 12 \sum_{k=1}^{t} \eta_k \sqrt{(1-\beta_{2,k})\mathcal{G}_k} \left\| \frac{\boldsymbol{G}_k}{\sqrt{\boldsymbol{W}_k}} \right\|_F^2 \\
&\leq \frac{1}{4} \sum_{k=1}^{t} \eta_k \left\| \frac{\bar{\boldsymbol{G}}_k}{\sqrt[4]{\boldsymbol{A}_k}} \right\|_F^2 + \frac{48\rho_0}{mn\epsilon_1^2} \sum_{k=1}^{t} \frac{\Sigma_k^2 \mathcal{G}_k^{3/2}}{k} \\
&\leq \frac{1}{4} \sum_{k=1}^{t} \rho_k \left\| \frac{\bar{\boldsymbol{G}}_k}{\sqrt[4]{\boldsymbol{A}_k}} \right\|_F^2 + \frac{48\rho_0 \Sigma_t^2 \mathcal{G}_t^{3/2}(1+\log t)}{mn\epsilon_1^2},
\end{aligned} \quad (59)$$

where we apply $\Sigma_k \leq \Sigma_t, \mathcal{G}_k \leq \mathcal{G}_t, k \leq t$ and Lemma A.1 in the last inequality.

**Estimating C.** Using the setups of $\eta_k$ and $\beta_{2,k}$, Lemma B.4 and Lemma A.1, we have

$$\mathbf{C} \leq \frac{L}{2} \sum_{k=1}^{t} \frac{\rho_0^2}{k} \left\| \frac{\boldsymbol{G}_k}{\sqrt{\boldsymbol{W}_k}} \right\|_F^2 \leq \frac{2L\rho_0^2}{mn\epsilon_1^2} \sum_{k=1}^{t} \frac{\Sigma_k^2 \mathcal{G}_k}{k} \leq \frac{2L\rho_0^2 \Sigma_t^2 \mathcal{G}_t(1+\log t)}{mn\epsilon_1^2}. \quad (60)$$

**An induction argument to bound $D_k$.** The induction is established based on the events in (42) and Lemma B.5. Hence, the target result will hold with probability at least $1 - 2\delta$. First, it's easy to verify that $G_1^2 \leq 2L(f(\boldsymbol{X}_1) - f^*) \leq H^2$ from Lemma A.2. Let us suppose that for some $t \in [T]$,

$$D_k \leq H, \quad \text{consequently with } \epsilon_1 = c_0/\sqrt{mn}, \quad \Sigma_k \leq \Sigma_H, \quad \mathcal{G}_k \leq \mathcal{H}, \quad \forall k \in [t], \quad (61)$$

where the specific defitions of $H, \Sigma_H$ and $\mathcal{H}$ are in (5). Then, we move to the case of $t+1$. We first subtract $f^*$ on both sides of (55) and use Lemma A.2 to derive that

$$\frac{\|\bar{\boldsymbol{G}}_{t+1}\|_F^2}{2L} \leq f(\boldsymbol{X}_{t+1}) - f^* \leq f(\boldsymbol{X}_1) - f^* + \mathbf{A} + \mathbf{B} + \mathbf{C}. \quad (62)$$

Next, we provide the esimtation for **A** based on (42) and Lemma B.5.

**Estimating A.** We first introduce $\boldsymbol{\xi}_k = \boldsymbol{G}_k - \bar{\boldsymbol{G}}_k$ into **A** and get that

$$\mathbf{A} = -\sum_{k=1}^{t} \eta_k \left\| \frac{\bar{\boldsymbol{G}}_k}{\sqrt[4]{\boldsymbol{A}_k}} \right\|_F^2 - \sum_{k=1}^{t} \eta_k \left\langle \bar{\boldsymbol{G}}_k, \frac{\boldsymbol{\xi}_k}{\sqrt{\boldsymbol{A}_k}} \right\rangle. \quad (63)$$

Under (42) and $\eta_k = \rho_k$, we can combinine with Lemma B.4 and $\rho_k$ in (7) to derive that when $c \in [0, 2)$,

$$\frac{\rho_k}{\sqrt{a_{ij}^{(k)}}} \leq \frac{\rho_0}{k^{1-c/2}} \cdot \frac{2\sqrt{\mathcal{G}_k}}{\sqrt{mn}\epsilon_1} \leq \frac{2\rho_0\sqrt{\mathcal{G}_k}}{\sqrt{mn}\epsilon_1}. \quad (64)$$

Therefore, setting $\lambda = \sqrt{mn}\epsilon_1/(6\sigma^2\rho_0\sqrt{\mathcal{H}})$ in (46), using (64) and re-scaling $\delta$, we derive that

$$-\sum_{k=1}^{t}\rho_k\left\langle\bar{G}_k,\frac{\xi_k}{\sqrt{A_k}}\right\rangle \leq \frac{1}{4}\sum_{k=1}^{t}\frac{\rho_k\sqrt{\mathcal{G}_k}}{\sqrt{\mathcal{H}}}\left\|\frac{\bar{G}_k}{\sqrt[4]{A_k}}\right\|_F^2 + \frac{6\sigma^2\rho_0\sqrt{\mathcal{H}}}{\sqrt{mn}\epsilon_1}\log\left(\frac{T}{\delta}\right), \quad \forall t \in [T],$$

which leads to that

$$\mathbf{A} \leq \sum_{k=1}^{t}\left(\frac{\sqrt{\mathcal{G}_k}}{4\sqrt{\mathcal{H}}}-1\right)\rho_k\left\|\frac{\bar{G}_k}{\sqrt[4]{A_k}}\right\|_F^2 + \frac{6\sigma^2\rho_0\sqrt{\mathcal{H}}}{\sqrt{mn}\epsilon_1}\log\left(\frac{T}{\delta}\right), \quad \forall t \in [T]. \tag{65}$$

**Putting together.** Note that the estimations in (59), (60) and (65) are established based on the probability events in (42) and Lemma B.5. Then, using (61), $\rho_0$ defined in (6) and $\epsilon_1 = \frac{c_0}{\sqrt{mn}}$ into these estimations, we have

$$\mathbf{A} \leq -\frac{3}{4}\sum_{k=1}^{t}\rho_k\left\|\frac{\bar{G}_k}{\sqrt[4]{A_k}}\right\|_F^2 + \frac{6\sigma^2\lambda_0}{Lc_0}\log\left(\frac{T}{\delta}\right),$$

$$\mathbf{B} \leq \frac{1}{4}\sum_{k=1}^{t}\rho_k\left\|\frac{\bar{G}_k}{\sqrt[4]{A_k}}\right\|_F^2 + \frac{48\lambda_0(1+\log T)}{Lc_0^2},$$

$$\mathbf{C} \leq \frac{2L\rho_0^2\Sigma_H^2\mathcal{H}(1+\log t)}{mn\epsilon_1^2} \leq \frac{2\lambda_0^2(1+\log T)}{Lc_0^2}. \tag{66}$$

Then, plugging (66) into (62), it leads to that

$$\frac{\|\bar{G}_{t+1}\|_F^2}{2L} \leq f(X_1) - f^* - \frac{1}{2}\sum_{k=1}^{t}\rho_k\left\|\frac{\bar{G}_k}{\sqrt[4]{A_k}}\right\|_F^2 + \frac{6\sigma^2\lambda_0}{Lc_0}\log\left(\frac{T}{\delta}\right)$$

$$+ \frac{2\lambda_0(24+\lambda_0)(1+\log T)}{Lc_0^2}. \tag{67}$$

With both sides multiplying $2L$, we derive that

$$\|\bar{G}_{t+1}\|_F^2 \leq 2L(f(X_1) - f^*) - L\sum_{k=1}^{t}\rho_k\left\|\frac{\bar{G}_k}{\sqrt[4]{A_k}}\right\|_F^2 + \frac{12\sigma^2\lambda_0}{c_0}\log\left(\frac{T}{\delta}\right)$$

$$+ \frac{4\lambda_0(24+\lambda_0)(1+\log T)}{c_0^2}$$

$$\leq H^2 - L\sum_{k=1}^{t}\rho_k\left\|\frac{\bar{G}_k}{\sqrt[4]{A_k}}\right\|_F^2 \leq H^2, \tag{68}$$

where $H$ is defined in (5). The induction is complete, and we prove the desired result. $\square$

## B.4 Proof of Theorem 6.1

The final convergence bound is established based on the probabilistic events in (42) and Lemma B.5, which thereby holds with probability at least $1 - 2\delta$. As a consequence of (68),

$$L\sum_{k=1}^{T}\rho_k\left\|\frac{\bar{G}_k}{\sqrt[4]{A_k}}\right\|_F^2 \leq H^2 - \|\bar{G}_{T+1}\|_F^2 \leq H^2. \tag{69}$$

Moreover, using Lemma B.4, Proposition B.1 and $\epsilon_1 = c_0/\sqrt{mn}$, we have

$$\sqrt{a_{ij}^{(k)}} \leq \sqrt{\Sigma_k^2 + \sqrt{mn}\epsilon_1} \leq \Sigma_H + \sqrt{c_0}, \quad \forall k \in [T]. \tag{70}$$

Thereby, with $\rho_k = \rho_0/k^{1-c/2}$, we have

$$\sum_{k=1}^{T}\rho_k\left\|\frac{\bar{G}_k}{\sqrt[4]{A_k}}\right\|_F^2 \geq \sum_{k=1}^{T}\frac{\rho_k\|\bar{G}_k\|_F^2}{\max_{i,j}\sqrt{a_{ij}^{(k)}}} \geq \frac{\rho_0}{\Sigma_H + \sqrt{c_0}}\sum_{k=1}^{T}\frac{\|\bar{G}_k\|_F^2}{k^{1-c/2}}. \tag{71}$$

Combining with (71) and (69), and using $\sum_{k=1}^{T}1/k^{1-c/2} \geq T^{c/2}$, we derive that

$$\min_{k\in[T]}\|\bar{G}_k\|_F^2 \leq \frac{H^2\left(\Sigma_H + \sqrt{c_0}\right)}{\rho_0 LT^{c/2}}.$$

## B.5 Proof of Corollary 1

Here, we let $\rho_k = \rho_0/\sqrt{T}, k \in [T]$, $\beta_{2,1} = 1/2$ and $\beta_{2,k} = \beta_2 = 1 - 1/T, k = 2, \cdots, T$ be a constant. We still suppose that the probability events in (42) and Lemma B.5 hold. Then, all the lemmas in Section B.2 still hold as they only require $\beta_{2,1} = 1/2, \beta_{2,k} \in [0, 1)$. Also, the estimation for $\mathbf{A}$ in (65) remains unchanged. Following the similar deduction in (58) and applying $\beta_{2,1} = 1/2$, $\beta_{2,k} = \beta_2 = 1 - 1/T, k \geq 2$ and $\rho_k = \rho_0/\sqrt{T}$, we have

$$
\begin{aligned}
\mathbf{B} &\leq \frac{1}{4} \sum_{k=1}^{t} \rho_k \left\| \frac{\bar{\boldsymbol{G}}_k}{\sqrt[4]{\boldsymbol{A}_k}} \right\|_F^2 + \frac{48\rho_0 \Sigma_t^2 \mathcal{G}_t^{3/2}}{mn\epsilon_1^2} \left( \sqrt{\frac{1}{2T}} + \sum_{k=2}^{t} \frac{1}{T} \right) \\
&\leq \frac{1}{4} \sum_{k=1}^{t} \rho_k \left\| \frac{\bar{\boldsymbol{G}}_k}{\sqrt[4]{\boldsymbol{A}_k}} \right\|_F^2 + \frac{96\rho_0 \Sigma_t^2 \mathcal{G}_t^{3/2}}{mn\epsilon_1^2}.
\end{aligned}
\tag{72}
$$

Following the similar deduction in (60), and using $\rho_k = \rho_0/\sqrt{T}$,

$$
\mathbf{C} \leq \frac{2L\rho_0^2}{mn\epsilon_1^2} \sum_{k=1}^{t} \frac{\Sigma_k^2 \mathcal{G}_k}{T} \leq \frac{2L\rho_0^2 \Sigma_t^2 \mathcal{G}_t}{mn\epsilon_1^2}.
\tag{73}
$$

Thereby, with the similar induction argument based on (42) and Lemma B.5, we can derive that with proabability at least $1 - 2\delta$, (69) and the following results hold

$$
D_t = \max_{k \in [t]} \|\bar{\boldsymbol{G}}_k\|_F \leq H, \quad \Sigma_t \leq \Sigma_H, \quad \mathcal{G}_t \leq \mathcal{H}, \quad \forall t \in [T],
\tag{74}
$$

when $H, \mathcal{H}$ and $\Sigma_H$ are as in (5) and

$$
0 < \rho_0 \leq \frac{\lambda_0}{L} \min \left\{ \frac{1}{\sqrt{\mathcal{H}}}, \frac{1}{2\Sigma_H^2 \mathcal{H}^{3/2}}, \frac{1}{\Sigma_H \sqrt{\mathcal{H}}} \right\}.
\tag{75}
$$

Hence, we will derive the convergence rate based on the probabilistic events in (42) and Lemma B.5, which thereby holds with probability at least $1 - 2\delta$. Since $\beta_{2,1} = 1/2$, $\beta_{2,k} \in [0, 1)$, $\epsilon_1 = c_0/\sqrt{mn}$ and (74) holds, we can get that

$$
\sqrt{a_{ij}^{(k)}} \leq \Sigma_H + \sqrt{c_0}, \quad \forall k \in [T].
$$

Following the same result in (71), and using $\rho_k = \rho_0/\sqrt{T}$, we have for any $k \in [T]$,

$$
\sum_{k=1}^{T} \rho_k \left\| \frac{\bar{\boldsymbol{G}}_k}{\sqrt[4]{\boldsymbol{A}_k}} \right\|_F^2 \geq \sum_{k=1}^{T} \frac{\rho_k \|\bar{\boldsymbol{G}}_k\|_F^2}{\max_{i,j} \sqrt{a_{ij}^{(k)}}} \geq \frac{\rho_0}{\Sigma_H + \sqrt{c_0}} \sum_{k=1}^{T} \frac{\|\bar{\boldsymbol{G}}_k\|_F^2}{\sqrt{T}}.
\tag{76}
$$

Then, combining (69), we get the desired result that

$$
\frac{1}{T} \sum_{k=1}^{T} \|\bar{\boldsymbol{G}}_k\|_F^2 \leq \frac{H^2 \left( \Sigma_H + \sqrt{c_0} \right)}{\rho_0 L \sqrt{T}}.
$$

# C  Proof detail for stochastic Adafactor with update clipping

## C.1  Proof preliminary

We follow the notation definitions of $D_k, \Sigma_k$ and $\mathcal{G}_k$ in (40). Next, we define

$$
\tilde{\boldsymbol{G}}_k = \frac{\boldsymbol{G}_k}{\max\{1, \|\boldsymbol{U}_k\|_F/(d_k\sqrt{mn})\}}.
\tag{77}
$$

Since $\text{RMS}(\boldsymbol{U}_k) = \|\boldsymbol{U}_k\|_F/\sqrt{mn}$, the update rule for Adafactor becomes

$$
\boldsymbol{X}_{k+1} = \boldsymbol{X}_k - \rho_k \frac{\tilde{\boldsymbol{G}}_k}{\sqrt{\boldsymbol{W}_k}}.
\tag{78}
$$

## C.2 Bounding gradient magnitude

Before proving the main convergence result, we still need to control the gradient magnitude through an induction argument in the following proposition. The proof detail, however, is different from the one for Proposition B.1. We will rely on some techniques in the analysis of algorithms with standrad clipping.

**Proposition C.1.** *Following the conditions and notations of Theorem 7.1, it holds that with probability at least $1 - 2\delta$,*

$$D_k \leq I, \quad \Sigma_k \leq \Sigma_I, \quad \mathcal{G}_k \leq \mathcal{I}, \quad \forall k \in [T].$$

*Proof.* Using the inequality in Lemma A.2 and (78), we have

$$f(\boldsymbol{X}_{k+1}) \leq f(\boldsymbol{X}_k) + \langle \bar{\boldsymbol{G}}_k, \boldsymbol{X}_{k+1} - \boldsymbol{X}_k \rangle + \frac{L}{2} \|\boldsymbol{X}_{k+1} - \boldsymbol{X}_k\|_F^2$$

$$= f(\boldsymbol{X}_k) - \rho_k \left\langle \bar{\boldsymbol{G}}_k, \frac{\tilde{\boldsymbol{G}}_k}{\sqrt{\boldsymbol{W}_k}} \right\rangle + \frac{L\rho_k^2}{2} \left\| \frac{\tilde{\boldsymbol{G}}_k}{\sqrt{\boldsymbol{W}_k}} \right\|_F^2.$$

Subtracting $f^*$ on both sides and summing up both sides over $k \in [t]$, we have for any $t \geq 1$,

$$f(\boldsymbol{X}_{t+1}) - f^* \leq f(\boldsymbol{X}_1) - f^* + \underbrace{\sum_{k=1}^{t} -\rho_k \left\langle \bar{\boldsymbol{G}}_k, \frac{\tilde{\boldsymbol{G}}_k}{\sqrt{\boldsymbol{W}_k}} \right\rangle}_{\mathbf{D}} + \underbrace{\sum_{k=1}^{t} \frac{L\rho_k^2}{2} \left\| \frac{\tilde{\boldsymbol{G}}_k}{\sqrt{\boldsymbol{W}_k}} \right\|_F^2}_{\mathbf{E}}. \tag{79}$$

Introducing $\boldsymbol{A}_k$ defined in (41), we further have the following decomposition,

$$\mathbf{D} = -\sum_{k=1}^{t} \rho_k \left\langle \bar{\boldsymbol{G}}_k, \frac{\tilde{\boldsymbol{G}}_k}{\sqrt{\boldsymbol{A}_k}} \right\rangle + \underbrace{\sum_{k=1}^{t} \rho_k \left\langle \bar{\boldsymbol{G}}_k, \left( \frac{1}{\sqrt{\boldsymbol{A}_k}} - \frac{1}{\sqrt{\boldsymbol{W}_k}} \right) \odot \tilde{\boldsymbol{G}}_k \right\rangle}_{\mathbf{D.1}}$$

$$= -\sum_{k=1}^{t} \rho_k \left\| \frac{\bar{\boldsymbol{G}}_k}{\sqrt[4]{\boldsymbol{A}_k}} \right\|_F^2 + \mathbf{D.1}$$

$$\underbrace{-\sum_{k=1}^{t} \rho_k \left\langle \bar{\boldsymbol{G}}_k, \frac{\tilde{\boldsymbol{G}}_k}{\sqrt{\boldsymbol{A}_k}} - \mathbb{E}_{\boldsymbol{Z}_k} \left[ \frac{\tilde{\boldsymbol{G}}_k}{\sqrt{\boldsymbol{A}_k}} \right] \right\rangle}_{\mathbf{D.2}} + \underbrace{\sum_{k=1}^{t} \rho_k \left\langle \bar{\boldsymbol{G}}_k, \frac{\bar{\boldsymbol{G}}_k}{\sqrt{\boldsymbol{A}_k}} - \mathbb{E}_{\boldsymbol{Z}_k} \left[ \frac{\tilde{\boldsymbol{G}}_k}{\sqrt{\boldsymbol{A}_k}} \right] \right\rangle}_{\mathbf{D.3}}. \tag{80}$$

In the following estimations, **D.1**, **E** and **D.3** are established based on the probability event in (42), whereas **D.2** does not rely on (42). First, based on (42), (77) and $D_k, \Sigma_k$ defined in (40), we have

$$\|\bar{\boldsymbol{G}}_k\|_F \leq D_k, \quad \|\boldsymbol{G}_k\|_F \leq \|\bar{\boldsymbol{G}}_k\|_F + \|\boldsymbol{G}_k - \bar{\boldsymbol{G}}_k\|_F \leq \Sigma_k, \quad \|\tilde{\boldsymbol{G}}_k\|_F \leq \|\boldsymbol{G}_k\|_F \leq \Sigma_k. \tag{81}$$

Hence, under (42), we get that

$$\|\mathbb{E}_{\boldsymbol{Z}_k}[\tilde{\boldsymbol{G}}_k]\|_F \leq \mathbb{E}_{\boldsymbol{Z}_k} \|\tilde{\boldsymbol{G}}_k\|_F \leq \mathbb{E}_{\boldsymbol{Z}_k} \|\boldsymbol{G}_k\|_F$$

$$\leq \mathbb{E}_{\boldsymbol{Z}_k} \|\bar{\boldsymbol{G}}_k\|_F + \mathbb{E}_{\boldsymbol{Z}_k} \|\boldsymbol{G}_k - \bar{\boldsymbol{G}}_k\|_F \leq D_k + \sigma \sqrt{\log\left(\frac{eT}{\delta}\right)} \leq \Sigma_k. \tag{82}$$

**Estimating E.** Under (42), we can use $\tilde{\boldsymbol{G}}_k$ defined in (77), Lemma B.4 and (81) to verify that

$$\left\| \frac{\tilde{\boldsymbol{G}}_k}{\sqrt{\boldsymbol{W}_k}} \right\|_F^2 \leq \frac{\|\tilde{\boldsymbol{G}}_k\|_F^2}{\min_{i,j} w_{ij}^{(k)}} \leq \frac{\|\boldsymbol{G}_k\|_F^2}{\min_{i,j} w_{ij}^{(k)}} \leq \frac{4\Sigma_k^2 \mathcal{G}_k}{mn\epsilon_1^2}. \tag{83}$$

Using $\rho_k = \rho_0/k^{1-c/2} \leq \rho_0/\sqrt{k}, \Sigma_k \leq \Sigma_t, \mathcal{G}_k \leq \mathcal{G}_t, \forall k \leq t$ and (83), we derive that

$$\mathbf{E} \leq \frac{L\rho_0^2}{2} \sum_{k=1}^{t} \frac{1}{k} \frac{4\Sigma_k^2 \mathcal{G}_k}{mn\epsilon_1^2} \leq \frac{2L\rho_0^2 \Sigma_t^2 \mathcal{G}_t}{mn\epsilon_1^2} \sum_{k=1}^{t} \frac{1}{k} \leq \frac{2L\rho_0^2 \Sigma_t^2 \mathcal{G}_t (1 + \log t)}{mn\epsilon_1^2}, \tag{84}$$

where the last inequality applies Lemma A.1.

**Estimating D.1** We can follow the similar deduction in (57) and (58) to derive that

$$
\mathbf{D.1} \leq \sum_{k=1}^{t} \sum_{i=1}^{n} \sum_{j=1}^{m} \rho_k \left| \bar{g}_{ij}^{(k)} \tilde{g}_{ij}^{(k)} \right| \left| \frac{1}{\sqrt{w_{ij}^{(k)}}} - \frac{1}{\sqrt{a_{ij}^{(k)}}} \right|
$$

$$
\leq \sum_{k=1}^{t} \sum_{i=1}^{n} \sum_{j=1}^{m} \rho_k \frac{\left| \bar{g}_{ij}^{(k)} \tilde{g}_{ij}^{(k)} \right|}{\sqrt{w_{ij}^{(k)}} \sqrt{a_{ij}^{(k)}}} \sqrt{\left| w_{ij}^{(k)} - a_{ij}^{(k)} \right|}
$$

$$
\leq \frac{1}{4} \sum_{k=1}^{t} \sum_{i=1}^{n} \sum_{j=1}^{m} \rho_k \cdot \frac{\left( \bar{g}_{ij}^{(k)} \right)^2}{\sqrt{a_{ij}^{(k)}}} + 4 \sum_{k=1}^{t} \sum_{i=1}^{n} \sum_{j=1}^{m} \rho_k \cdot \frac{\left| w_{ij}^{(k)} - a_{ij}^{(k)} \right|}{\sqrt{a_{ij}^{(k)}}} \cdot \left( \frac{\tilde{g}_{ij}^{(k)}}{\sqrt{w_{ij}^{(k)}}} \right)^2 . \tag{85}
$$

Under (42), we can apply Lemma B.6, $\mathcal{G}_k \leq \mathcal{G}_t, \forall k \leq t$, and (83) into (85) to derive that

$$
\mathbf{D.1} \leq \frac{1}{4} \sum_{k=1}^{t} \rho_k \left\| \frac{\bar{G}_k}{\sqrt[4]{A_k}} \right\|_F^2 + 12 \sum_{k=1}^{t} \rho_k \sqrt{(1 - \beta_{2,k}) \mathcal{G}_k} \left\| \frac{\tilde{G}_k}{\sqrt{W_k}} \right\|_F^2
$$

$$
\leq \frac{1}{4} \sum_{k=1}^{t} \rho_k \left\| \frac{\bar{G}_k}{\sqrt[4]{A_k}} \right\|_F^2 + \frac{48 \Sigma_t^2 \mathcal{G}_t^{3/2}}{mn\epsilon_1^2} \sum_{k=1}^{t} \rho_k \sqrt{1 - \beta_{2,k}} . \tag{86}
$$

Using $\rho_k = \rho_0 / k^{1-c/2}, \beta_{2,k} = 1 - 1/k^c$ and Lemma A.1, we further have

$$
\mathbf{D.1} \leq \frac{1}{4} \sum_{k=1}^{t} \rho_k \left\| \frac{\bar{G}_k}{\sqrt[4]{A_k}} \right\|_F^2 + \frac{48 \rho_0 \Sigma_t^2 \mathcal{G}_t^{3/2} (1 + \log t)}{mn\epsilon_1^2} . \tag{87}
$$

**Estimating D.2.** Since $A_k$ is independent from $Z_k$, it leads to

$$
\mathbf{D.2} = -\sum_{k=1}^{t} \rho_k \left\langle \frac{\bar{G}_k}{\sqrt{A_k}}, \tilde{G}_k - \mathbb{E}_{Z_k} \left[ \tilde{G}_k \right] \right\rangle .
$$

Let $\varphi_k := -\rho_k \left\langle \frac{\bar{G}_k}{\sqrt{A_k}}, \tilde{G}_k - \mathbb{E}_{Z_k} \left[ \tilde{G}_k \right] \right\rangle$ and the filtration $\mathcal{F}_k := \sigma(Z_1, \cdots, Z_k)$. Note that $\rho_k$, $\bar{G}_k$ and $A_k$ are measurable with $\mathcal{F}_{k-1}$. Since $\xi_k$ is measurable with $\mathcal{F}_k$, we could prove that $\{\varphi_k\}_{k \geq 1}$ is a martingale difference sequence by showing that

$$
\mathbb{E}[\varphi_k \mid \mathcal{F}_{k-1}] = -\rho_k \left\langle \frac{\bar{G}_k}{\sqrt{A_k}}, \mathbb{E}_{Z_k} \left[ \tilde{G}_k - \mathbb{E}_{Z_k} [\tilde{G}_k] \right] \right\rangle = 0 .
$$

Using that $\|\tilde{G}_k\|_F \leq \|G_k\|_F$ and $\|\bar{G}_k\|_F \leq D_k$, we derive that

$$
\|\tilde{G}_k - \mathbb{E}_{Z_k}[\tilde{G}_k]\|_F^2 \leq (\|G_k\|_F + \mathbb{E}_{Z_k} \|G_k\|_F)^2
$$

$$
\leq \left( \|G_k - \bar{G}_k\|_F + \|\bar{G}_k\|_F + \mathbb{E}_{Z_k} \|G_k - \bar{G}_k\|_F + \mathbb{E}_{Z_k} \|\bar{G}_k\|_F \right)^2
$$

$$
\leq \left( \|G_k - \bar{G}_k\|_F + \mathbb{E}_{Z_k} \|G_k - \bar{G}_k\|_F + 2D_k \right)^2 . \tag{88}
$$

Let $\omega_k' = 4\Sigma_k \rho_k \left\| \frac{\bar{G}_k}{\sqrt{A_k}} \right\|_F$ which is measurable with $\mathcal{F}_{k-1}$. We thus derive from the Cauchy-Schwarz inequality, (88) and $\sigma \leq \Sigma_k, D_k \leq \Sigma_k$,

$$
\mathbb{E} \left[ \exp \left( \frac{\varphi_k^2}{(\omega_k')^2} \right) \mid \mathcal{F}_{k-1} \right] \leq \mathbb{E} \left[ \exp \left( \frac{\left\| \frac{\bar{G}_k}{\sqrt{A_k}} \right\|_F^2 \|\tilde{G}_k - \mathbb{E}_{Z_k}[\tilde{G}_k]\|_F^2}{\left\| \frac{\bar{G}_k}{\sqrt{A_k}} \right\|_F^2 \cdot 16\Sigma_k^2} \right) \left| \mathcal{F}_{k-1} \right. \right]
$$

$$
\leq \mathbb{E} \left[ \exp \left( \frac{2(\|G_k - \bar{G}_k\|_F + \mathbb{E}_{Z_k} \|G_k - \bar{G}_k\|_F)^2 + 8D_k^2}{16\Sigma_k^2} \right) \left| \mathcal{F}_{k-1} \right. \right]
$$

$$
\leq \mathbb{E} \left[ \exp \left( \frac{\|G_k - \bar{G}_k\|_F^2 + (\mathbb{E}_{Z_k} \|G_k - \bar{G}_k\|_F)^2}{4\sigma^2} \right) \left| \mathcal{F}_{k-1} \right. \right] \cdot \exp(1/2)
$$

$$
\leq \mathbb{E} \left[ \exp \left( \frac{\|G_k - \bar{G}_k\|_F^2 + \mathbb{E}_{Z_k} \|G_k - \bar{G}_k\|_F^2}{4\sigma^2} \right) \left| \mathcal{F}_{k-1} \right. \right] \cdot \exp(1/2) .
$$

Using Jensen's inequality, we get that

$$\mathbb{E}\left[\exp\left(\frac{\mathbb{E}_{\boldsymbol{Z}_k}\|\boldsymbol{G}_k - \bar{\boldsymbol{G}}_k\|_F^2}{4\sigma^2}\right)\Big|\mathcal{F}_{k-1}\right] \leq \mathbb{E}\left[\mathbb{E}_{\boldsymbol{Z}_k}\left(\exp\left(\frac{\|\boldsymbol{G}_k - \bar{\boldsymbol{G}}_k\|_F^2}{4\sigma^2}\right)\right)\Big|\mathcal{F}_{k-1}\right]$$

$$= \mathbb{E}\left[\exp\left(\frac{\|\boldsymbol{G}_k - \bar{\boldsymbol{G}}_k\|_F^2}{4\sigma^2}\right)\Big|\mathcal{F}_{k-1}\right].$$

Using Jensen's inequality and the definition of sub-Gaussian noise, we further have

$$\mathbb{E}\left[\exp\left(\frac{\|\boldsymbol{G}_k - \bar{\boldsymbol{G}}_k\|_F^2}{4\sigma^2}\right)\Big|\mathcal{F}_{k-1}\right] \leq \left(\mathbb{E}\left[\exp\left(\frac{\|\boldsymbol{G}_k - \bar{\boldsymbol{G}}_k\|_F^2}{\sigma^2}\right)\Big|\mathcal{F}_{k-1}\right]\right)^{1/4} \leq \mathrm{e}^{1/4}.$$

Combining the above, we get that $\mathbb{E}\left[\exp\left(\frac{\varphi_k^2}{(\omega_k')^2}\right)\mid\mathcal{F}_{k-1}\right] \leq \mathrm{e}$. Then, using Lemma B.1 and (42), it leads to that for any $\lambda > 0$, with probability at least $1-\delta$, for all $t \in [T]$,

$$\mathbf{D.2} = \sum_{k=1}^t \varphi_k \leq 12\lambda \sum_{k=1}^t \Sigma_k^2 \rho_k^2 \left\|\frac{\bar{\boldsymbol{G}}_k}{\sqrt{\boldsymbol{A}_k}}\right\|_F^2 + \frac{1}{\lambda}\log\left(\frac{T}{\delta}\right)$$

$$= 12\lambda \sum_{k=1}^t \sum_{i=1}^n \sum_{j=1}^m \frac{\rho_k \Sigma_k^2}{\sqrt{a_{ij}^{(k)}}} \cdot \rho_k \frac{\left(\bar{g}_{ij}^{(k)}\right)^2}{\sqrt{a_{ij}^{(k)}}} + \frac{1}{\lambda}\log\left(\frac{T}{\delta}\right). \tag{89}$$

**Estimating D.3.** First, since $\boldsymbol{A}_k$ is independent from $\boldsymbol{Z}_k$ and $\mathbb{E}_{\boldsymbol{Z}_k}[\boldsymbol{G}_k] = \bar{\boldsymbol{G}}_k$, we have

$$\mathbf{D.3} = \sum_{k=1}^t \rho_k \left\langle \bar{\boldsymbol{G}}_k, \frac{\mathbb{E}_{\boldsymbol{Z}_k}[\boldsymbol{G}_k]}{\sqrt{\boldsymbol{A}_k}} - \frac{\mathbb{E}_{\boldsymbol{Z}_k}[\tilde{\boldsymbol{G}}_k]}{\sqrt{\boldsymbol{A}_k}}\right\rangle$$

$$\leq \sum_{k=1}^t \rho_k \left\|\frac{\bar{\boldsymbol{G}}_k}{\sqrt{\boldsymbol{A}_k}}\right\|_F \cdot \left\|\mathbb{E}_{\boldsymbol{Z}_k}\underbrace{\left[\boldsymbol{G}_k - \frac{\boldsymbol{G}_k}{\max\{1, \|\boldsymbol{U}_k\|_F/(d_k\sqrt{mn})\}}\right]}_{\boldsymbol{\Omega}_k}\right\|_F. \tag{90}$$

Then, we will estimate $\mathbb{E}_{\boldsymbol{Z}_k}\boldsymbol{\Omega}_k$ under the event in (42) and consequently (14) that we restate here:

$$\max_{l\in[k]}\|\boldsymbol{G}_l\|_F \leq \Sigma_k, \forall k \in [T]. \tag{91}$$

We note that $\mathbb{E}_{\boldsymbol{Z}_k}\boldsymbol{\Omega}_k$ is a random variable depending only on $\{\boldsymbol{Z}_1, \cdots, \boldsymbol{Z}_{k-1}\}$ and $\boldsymbol{Z}_k$ can be replaced by any $\boldsymbol{Z}_k'$ that is i.i.d. with $\boldsymbol{Z}_1, \cdots, \boldsymbol{Z}_{k-1}$ and we shall use the similar notations such as $\boldsymbol{\xi}_k'$, $\boldsymbol{\Omega}'$ and $\boldsymbol{U}_k'$ for the corresponding variables with $\boldsymbol{Z}_k$ replaced by $\boldsymbol{Z}_k'$. Then, we define the indicator functions $\hat{S}_{k,1}$ and $\hat{S}_{k,2}$ as follows,

$$\hat{S}_{k,1} = \chi_{\left\{\|\boldsymbol{\xi}_k'\|_F^2 \leq \sigma^2 \log\left(\frac{\mathrm{e}T}{\delta}\right)\right\}}, \quad \hat{S}_{k,2} = \chi_{\left\{\|\boldsymbol{\xi}_k'\|_F^2 > \sigma^2 \log\left(\frac{\mathrm{e}T}{\delta}\right)\right\}}.$$

Using Hölder's inequality and (82), we derive that,

$$\mathbb{E}_{\boldsymbol{Z}_k'}\left[\|\boldsymbol{\Omega}'_k\|_F \hat{S}_{k,2}\right] \leq \sqrt{\mathbb{E}_{\boldsymbol{Z}_k'}\|\boldsymbol{\Omega}'_k\|_F^2} \cdot \sqrt{\mathbb{E}_{\boldsymbol{Z}_k'}[\hat{S}_{k,2}^2]}$$

$$\leq \sqrt{\mathbb{E}_{\boldsymbol{Z}_k'}\|\boldsymbol{G}_k'\|_F^2} \cdot \sqrt{\mathbb{E}_{\boldsymbol{Z}_k'}[\hat{S}_{k,2}^2]} \leq \Sigma_k\sqrt{\frac{\delta}{T}},$$

where the last inequality uses the following result since (42) holds, $\hat{S}_{k,2}$ is dependent from $\boldsymbol{Z}_1, \cdots, \boldsymbol{Z}_k'$ and $\boldsymbol{Z}_k'$ is independent from $\boldsymbol{Z}_1, \cdots, \boldsymbol{Z}_{k-1}$,

$$\mathbb{E}_{\boldsymbol{Z}_k'}[\hat{S}_{k,2}^2] = \mathbb{P}\left(\|\boldsymbol{\xi}_k'\|_F^2 > \sigma^2 \log\left(\frac{\mathrm{e}T}{\delta}\right)\mid\boldsymbol{Z}_1, \cdots \boldsymbol{Z}_{k-1}\right) \leq \frac{\delta}{T}.$$

We next define the indicator functions $S_{k,1}$, $S_{k,2}$ and $\tilde{S}_{k,1}$ as follows,

$$S_{k,1} = \chi_{\{\|\boldsymbol{U}_k'\|_F \geq d_k\sqrt{mn}\}}\hat{S}_{k,1}, \quad S_{k,2} = \chi_{\{\|\boldsymbol{U}_k'\|_F < d_k\sqrt{mn}\}}\hat{S}_{k,1}, \quad \tilde{S}_{k,1} = \chi_{\left\{\|\boldsymbol{G}_k'\|_F \geq \frac{d_k mn\epsilon_1}{2\sqrt{\mathcal{G}_k}}\right\}}\hat{S}_{k,1}.$$

Under (91) and the event of $\hat{S}_{k,1}$, we can use the similar deduction in Lemma B.4 to derive that

$$\|\boldsymbol{U}_k'\|_F = \left\| \frac{\boldsymbol{G}_k'}{\sqrt{\boldsymbol{W}_k'}} \right\|_F \leq \frac{\|\boldsymbol{G}_k'\|_F}{\min_{i,j} \sqrt{(w_{ij}^{(k)})'}} \leq \|\boldsymbol{G}_k'\|_F \cdot \frac{2\sqrt{\mathcal{G}_k}}{\sqrt{mn}\epsilon_1}, \quad \|\boldsymbol{G}_k'\|_F \leq \Sigma_k. \qquad (92)$$

Consequently, we have $S_{k,1} \leq \tilde{S}_{k,1}$ from (92). Note that when $S_{k,2} = 1$, it implies that $\boldsymbol{\Omega}_k' = \boldsymbol{0}_{n\times m}$. Then, we derive that

$$\left\| \mathbb{E}_{\boldsymbol{Z}_k'}[\boldsymbol{\Omega}_k' \hat{S}_{k,1}] \right\|_F = \left\| \mathbb{E}_{\boldsymbol{Z}_k'}[\boldsymbol{\Omega}_k' S_{k,1}] + \mathbb{E}_{\boldsymbol{Z}_k'}[\boldsymbol{\Omega}_k' S_{k,2}] \right\|_F = \left\| \mathbb{E}_{\boldsymbol{Z}_k'}[\boldsymbol{\Omega}_k' S_{k,1}] \right\|_F$$

$$\leq \mathbb{E}_{\boldsymbol{Z}_k'}[S_{k,1} \|\boldsymbol{\Omega}_k'\|_F] \leq \mathbb{E}_{\boldsymbol{Z}_k'}\left[ \tilde{S}_{k,1} \|\boldsymbol{\Omega}_k'\|_F \right] \leq \mathbb{E}_{\boldsymbol{Z}_k'}\left[ \tilde{S}_{k,1} \|\boldsymbol{G}_k'\|_F \right],$$

where the last inequality applies $\|\boldsymbol{\Omega}_k'\|_F \leq \|\boldsymbol{G}_k'\|_F$ from (90). Note that when $\tilde{S}_{k,1} = 1$, $\frac{d_k mn\epsilon_1}{2\sqrt{\mathcal{G}_k}} \leq \|\boldsymbol{G}_k'\|_F \leq \Sigma_k$. Using that $\mathcal{G}_k$ and $\Sigma_k$ are indepedent from $\boldsymbol{Z}_k'$, and noting that $\alpha > 1$, we have

$$\mathbb{E}_{\boldsymbol{Z}_k'}\left[ \tilde{S}_{k,1} \|\boldsymbol{G}_k'\|_F \right] \leq \mathbb{E}_{\boldsymbol{Z}_k'}\left[ \tilde{S}_{k,1} \|\boldsymbol{G}_k'\|_F^\alpha \|\boldsymbol{G}_k'\|_F^{1-\alpha} \right] \leq \Sigma_k^\alpha \left( \frac{2\sqrt{\mathcal{G}_k}}{d_k mn\epsilon_1} \right)^{\alpha-1}.$$

From the above analysis, we derive that under (91),

$$\|\mathbb{E}_{\boldsymbol{Z}_k}[\boldsymbol{\Omega}_k]\|_F = \left\| \mathbb{E}_{\boldsymbol{Z}_k'}[\boldsymbol{\Omega}_k'] \right\|_F \leq \Sigma_k \sqrt{\frac{\delta}{T}} + \Sigma_k^\alpha \left( \frac{2\sqrt{\mathcal{G}_k}}{d_k mn\epsilon_1} \right)^{\alpha-1}. \qquad (93)$$

Under (91), we can use Lemma B.4 to get that,

$$\left\| \frac{\bar{\boldsymbol{G}}_k}{\sqrt{\boldsymbol{A}_k}} \right\|_F \leq \frac{\|\bar{\boldsymbol{G}}_k\|_F}{\min_{i,j} \sqrt{a_{ij}^{(k)}}} \leq \frac{2D_k \sqrt{\mathcal{G}_k}}{\sqrt{mn}\epsilon_1}. \qquad (94)$$

Combining with (90), (93) and (94), and using $\rho_k = \rho_0/k^{1-c/2}, c \in (0,1], d_k^{\alpha-1} \geq k^{c/2}$ and Lemma A.1, we derive that under (91),

$$\textbf{D.3} \leq \sum_{k=1}^t \frac{2\rho_k D_k \sqrt{\mathcal{G}_k}}{\sqrt{mn}\epsilon_1} \left( \Sigma_k \sqrt{\frac{\delta}{T}} + \Sigma_k^\alpha \left( \frac{2\sqrt{\mathcal{G}_k}}{d_k mn\epsilon_1} \right)^{\alpha-1} \right)$$

$$\leq \frac{2\rho_0 D_t \sqrt{\mathcal{G}_t}}{\sqrt{mn}\epsilon_1} \left( \Sigma_t \sqrt{\delta} + \Sigma_t^\alpha \left( \frac{2\sqrt{\mathcal{G}_t}}{mn\epsilon_1} \right)^{\alpha-1} \right) (1 + \log T), \qquad (95)$$

where the last inequality further uses $D_k \leq D_t, \Sigma_k \leq \Sigma_t, \mathcal{G}_k \leq \mathcal{G}_t$ when $k \leq t$.

**An induction argument.** The induction argument is based on the probability events in (42) and (89), thereby the desired result holds with probability at least $1 - 2\delta$. First, we can easily verify that $D_1^2 \leq 2L(f(\boldsymbol{X}_1) - f^*) \leq I^2$ from Lemma A.2. Let us suppose that for some $t \in [T]$,

$$D_k \leq I, \quad \text{consequently,} \quad \Sigma_k \leq \Sigma_I, \quad \mathcal{G}_k \leq \mathcal{I}, \quad \forall k \in [t]. \qquad (96)$$

Since (42) holds, we can first use $\rho_k = \rho_0/k^{1-c/2}, c \in [0,1]$, (42) and Lemma B.4 to derive that

$$\frac{\rho_k}{\sqrt{a_{ij}^{(k)}}} \leq \frac{\rho_0}{k^{1-c/2}} \cdot \frac{2\sqrt{\mathcal{G}_k}}{\sqrt{mn}\epsilon_1} \leq \frac{2\rho_0 \sqrt{\mathcal{G}_k}}{\sqrt{mn}\epsilon_1}. \qquad (97)$$

Plugging (97) into (89) and re-scaling $\delta$, it leads to that for any $\lambda > 0$, with probability at least $1 - 2\delta$,

$$\textbf{D.2} \leq \frac{24\lambda\rho_0}{\sqrt{mn}\epsilon_1} \sum_{k=1}^t \rho_k \Sigma_k^2 \sqrt{\mathcal{G}_k} \left\| \frac{\bar{\boldsymbol{G}}_k}{\sqrt[4]{\boldsymbol{A}_k}} \right\|_F^2 + \frac{1}{\lambda} \log \left( \frac{T}{\delta} \right), \quad \forall t \in [T].$$

Setting $\lambda = (\sqrt{mn}\epsilon_1)/(96\Sigma_I^2 \sqrt{\mathcal{I}}\rho_0)$ where $\Sigma_I, \mathcal{I}$ are as in Theorem 7.1, we then derive that

$$\textbf{D.2} \leq \frac{1}{4} \sum_{k=1}^t \frac{\rho_k \Sigma_k^2 \sqrt{\mathcal{G}_k}}{\Sigma_I^2 \sqrt{\mathcal{I}}} \left\| \frac{\bar{\boldsymbol{G}}_k}{\sqrt[4]{\boldsymbol{A}_k}} \right\|_F^2 + \frac{96\Sigma_I^2 \sqrt{\mathcal{I}}\rho_0}{\sqrt{mn}\epsilon_1} \log \left( \frac{T}{\delta} \right), \quad \forall t \in [T]. \qquad (98)$$

Then, we can plug the estimations (84), (87), (98) and (95) into (80) and (79), and use (96) to get that

$$
f(\boldsymbol{X}_{t+1}) - f^* \le f(\boldsymbol{X}_1) - f^* - \frac{1}{2}\sum_{k=1}^{t}\rho_k\left\|\frac{\bar{\boldsymbol{G}}_k}{\sqrt[4]{\boldsymbol{A}_k}}\right\|_F^2 + \frac{48\rho_0\Sigma_I^2\mathcal{I}^{3/2}(1+\log T)}{mn\epsilon_1^2}
$$

$$
+ \frac{96\rho_0\Sigma_I^2\sqrt{\mathcal{I}}}{\sqrt{mn}\epsilon_1}\log\left(\frac{T}{\delta}\right) + \frac{2\rho_0 I\sqrt{\mathcal{I}}}{\sqrt{mn}\epsilon_1}\left(\Sigma_I\sqrt{\delta} + \Sigma_I^\alpha\left(\frac{2\sqrt{\mathcal{I}}}{mn\epsilon_1}\right)^{\alpha-1}\right)(1+\log T)
$$

$$
+ \frac{2L\rho_0^2\Sigma_I^2\mathcal{I}(1+\log T)}{mn\epsilon_1^2}. \tag{99}
$$

Recalling the condition for $\rho_0$ in (10) and $\epsilon_1 = c_0/\sqrt{mn}$, then using $\|\bar{\boldsymbol{G}}_{t+1}\|_F^2 \le 2L(f(\boldsymbol{X}_{t+1}) - f^*)$ from Lemma A.2,

$$
\frac{\|\bar{\boldsymbol{G}}_{t+1}\|_F^2}{2L} \le f(\boldsymbol{X}_1) - f^* - \frac{1}{2}\sum_{k=1}^{t}\rho_k\left\|\frac{\bar{\boldsymbol{G}}_k}{\sqrt[4]{\boldsymbol{A}_k}}\right\|_F^2 + \frac{2\sqrt{\delta}(1+\log T)}{c_0}\cdot\frac{\lambda_0}{L}
$$

$$
+ \frac{(48\lambda_0 + 2\lambda_0^2)(1+\log T)}{Lc_0^2} + \frac{96\lambda_0}{Lc_0}\log\left(\frac{T}{\delta}\right) + \frac{2^\alpha\lambda_0(1+\log T)}{L(mn)^{(\alpha-1)/2}c_0^\alpha}.
$$

With both sides multiplying $2L$, we obtain that

$$
\|\bar{\boldsymbol{G}}_{t+1}\|_F^2 \le I^2 - L\sum_{k=1}^{t}\rho_k\left\|\frac{\bar{\boldsymbol{G}}_k}{\sqrt[4]{\boldsymbol{A}_k}}\right\|_F^2 \le I^2, \tag{100}
$$

where $I$ is defined in (9). Then, the induction is complete, and we prove the desired result. $\qquad\square$

## C.3 Proof of Theorem 7.1

The final convergence bound is established based on (42) and (89), which thereby holds with probability at least $1 - 2\delta$. As a consequence of (100),

$$
L\sum_{k=1}^{T}\rho_k\left\|\frac{\bar{\boldsymbol{G}}_k}{\sqrt[4]{\boldsymbol{A}_k}}\right\|_F^2 \le I^2 - \|\bar{\boldsymbol{G}}_{T+1}\|_F^2 \le I^2. \tag{101}
$$

Under (42), we can use Lemma B.4 and Proposition C.1 to get that $a_{ij}^{(k)} \le \Sigma_k^2 + \min\{m,n\}\epsilon_1 \le \Sigma_I^2 + \sqrt{mn}\epsilon_1$ for all $k \in [T]$. With $\epsilon_1 = c_0/\sqrt{mn}$ and $\rho_k = \rho_0/k^{1-c/2}$, we have

$$
\sum_{k=1}^{T}\rho_k\left\|\frac{\bar{\boldsymbol{G}}_k}{\sqrt[4]{\boldsymbol{A}_k}}\right\|_F^2 \ge \sum_{k=1}^{T}\frac{\rho_k\|\bar{\boldsymbol{G}}_k\|_F^2}{\max_{i,j}\sqrt{a_{ij}^{(k)}}} \ge \frac{\rho_0}{\sqrt{\Sigma_I^2 + c_0}}\sum_{k=1}^{T}\frac{\|\bar{\boldsymbol{G}}_k\|_F^2}{k^{1-c/2}}. \tag{102}
$$

Using $\sum_{k=1}^{T}1/k^{1-c/2} \ge T^{c/2}$, we derive that

$$
\min_{k\in[T]}\|\bar{\boldsymbol{G}}_k\|_F^2 \le \frac{I^2\sqrt{\Sigma_I^2 + c_0}}{\rho_0 LT^{c/2}} \le \frac{I^2}{\rho_0 LT^{c/2}}\left(I + \sigma\sqrt{\log\left(\frac{\mathrm{e}T}{\delta}\right)} + \sqrt{c_0}\right).
$$

## C.4 Proof of Corollary 2

Here, we let $\rho_k = \rho_0/\sqrt{T}, k \in [T]$, $\beta_{2,1} = 1/2$ and $\beta_{2,k} = \beta_2 = 1 - 1/T, k = 2,3,\cdots,T$. Setting $\beta_{2,k} = 1 - 1/T$ and $\rho_k = \rho_0/\sqrt{T}$, the estimations in (84), (98) and (95) remain unchanged under the probability event in (42). Indeed, these estimations can be further tighten by replacing $\sum_{k=1}^{t}\frac{1}{k} \le 1 + \log t$ with $\sum_{k=1}^{t}\frac{1}{T} \le 1$. The minor difference comes from the estimation of **D.1**. Following the similar deduction in (85) and (86), and using the new setups for $\rho_k$ and $\beta_{2,k}$, we have

$$
\textbf{D.1} \le \frac{1}{4}\sum_{k=1}^{t}\rho_k\left\|\frac{\bar{\boldsymbol{G}}_k}{\sqrt[4]{\boldsymbol{A}_k}}\right\|_F^2 + \frac{48\rho_0\Sigma_t^2\mathcal{G}_t^{3/2}}{mn\epsilon_1^2}\left(\sqrt{\frac{1}{2T}} + \sum_{k=2}^{t}\frac{1}{T}\right)
$$

$$
\le \frac{1}{4}\sum_{k=1}^{t}\rho_k\left\|\frac{\bar{\boldsymbol{G}}_k}{\sqrt[4]{\boldsymbol{A}_k}}\right\|_F^2 + \frac{96\rho_0\Sigma_t^2\mathcal{G}_t^{3/2}}{mn\epsilon_1^2}. \tag{103}
$$

Thereby, with the induction argument and the probability events in (42) and (89), we can still verify that with probability at least $1 - 2\delta$, (100) and the following inequalities hold

$$D_k \leq I, \quad \Sigma_k \leq \Sigma_I, \quad \mathcal{G}_k \leq \mathcal{I}, \quad \forall k \in [T], \tag{104}$$

when

$$0 < \rho_0 \leq \frac{\lambda_0}{L} \min \left\{ \frac{1}{\Sigma_I^2 \sqrt{\mathcal{I}}}, \frac{1}{2\Sigma_I^2 \mathcal{I}^{3/2}}, \frac{1}{\Sigma_I \sqrt{\mathcal{I}}}, \frac{1}{I(\Sigma_I \sqrt{\mathcal{I}})^\alpha} \right\}. \tag{105}$$

Hence, we establish the convergence rate based on the (42) and (89), which thereby holds with probability at least $1 - 2\delta$. Since $\beta_{2,1} = 1/2$, $\beta_{2,k} \in [0,1)$, $k \geq 2$ and $\epsilon_1 = c_0/\sqrt{mn}$, we can use Lemma B.4 and Proposition C.1 to get that

$$a_{ij}^{(k)} \leq \Sigma_k^2 + \min\{m,n\}\epsilon_1 \leq \Sigma_I^2 + \sqrt{mn}\epsilon_1, \quad \forall k \in [T].$$

Then, using $\rho_k = \rho_0/\sqrt{T}$, we have

$$\sum_{k=1}^T \rho_k \left\| \frac{\bar{G}_k}{\sqrt[4]{A_k}} \right\|_F^2 \geq \sum_{k=1}^T \frac{\rho_k \|\bar{G}_k\|_F^2}{\max_{i,j} \sqrt{a_{ij}^{(k)}}} \geq \frac{\rho_0}{\sqrt{\Sigma_I^2 + c_0}} \sum_{k=1}^T \frac{\|\bar{G}_k\|_F^2}{\sqrt{T}}. \tag{106}$$

Then, combining (101), we get the desired result that

$$\frac{1}{T} \sum_{k=1}^T \|\bar{G}_k\|_F^2 \leq \frac{I^2 \sqrt{\Sigma_I^2 + c_0}}{\rho_0 L \sqrt{T}} \leq \frac{I^2}{\rho_0 L \sqrt{T}} \left( I + \sigma \sqrt{\log\left(\frac{\mathrm{e}T}{\delta}\right)} + \sqrt{c_0} \right).$$

## D   Some complementary experiments

All the experiments are conducted using the fairseq implementation of Adafactor [4] and the Hugging Face implementation of Adam on two NVIDIA GeForce RTX 4090 GPUs. The pretrained models of BERT-Base/Large and GPT-2 are also downloaded from Hugging Face.

### D.1   Experiments on Adafactor without update clipping

We conduct experiments on BERT-Base and BERT-Large using the GLUE/MNLI benchmark, and on GPT-2 using the BookCorpus dataset. All models are trained with the Adafactor optimizer without update clipping, under the parameter setting $\beta_{2,k} = 1 - 1/k^c$ and $\rho_k = \rho_0/k^c$, where the decay rate $c$ ranges over $\{0.6, 0.7, 0.8, 0.9, 1.0\}$. Additionally, we compare the optimal performance under our setup (with $c = 1$) against both the default Adafactor configuration proposed by [38], that is, $\beta_{2,k} = 1 - 1/k^{0.8}$ and $\rho_k = \rho_0/\sqrt{k}$, and the Adam optimizer with $\beta_1 = 0.9, \beta_2 = 0.999$.

Each experiment is conducted over three epochs with a batch size of 128 for BERT-Base/Large and a batch size of 8 for GPT-2. The base learning rate $\rho_0$ is selected via a two-stage grid search. First, we search over the coarse grid $\{1, 0.1, 0.01, 0.001, 0.0001\}$. Then, based on the best candidate (e.g., 0.001), we refine the search by evaluating its surrounding values with a step-size equal to one-tenth of the candidate value (e.g., $1 \times 10^{-4}$), and choose the best-performing learning rate. All training loss curves and test accuracy results are presented in Figures 2, Figure 3, and Table 1.

Our results show that both convergence rates and test accuracy consistently improve as the decay rate $c$ increases from 0.6 to 1.0, with the best performance achieved at $c = 1$, which aligns well with Theorem 6.1. The training loss at $c = 1$ is slightly better or comparable to that under the default Adafactor setting. However, test accuracy is marginally worse, which may be attributed to overfitting under this configuration.

Furthermore, the best performances of Adafactor (at $c = 1$) for training BERT-Base and BERT-Large are comparable to that of Adam, suggesting that the reduced memory overhead in Adafactor does not necessarily compromise convergence speed or generalization performance.

---

[4]https://github.com/facebookresearch/fairseq/blob/main/fairseq/optim/adafactor.py

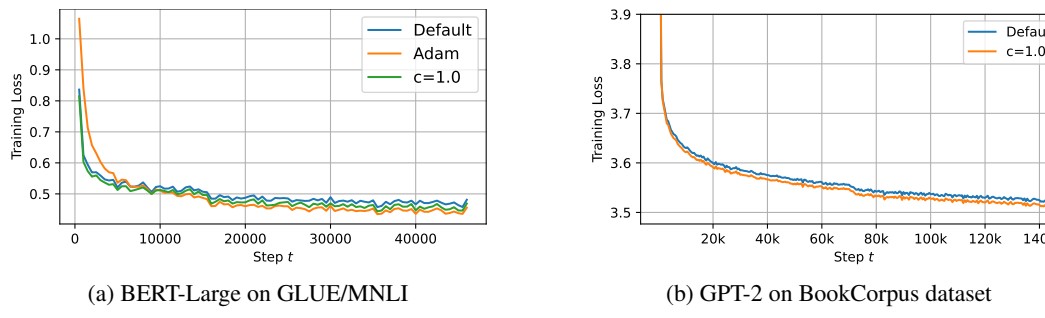

(a) BERT-Large on GLUE/MNLI

(b) GPT-2 on BookCorpus dataset

Figure 2: Training loss of Adafactor (no update clipping) with $c = 1$ or default setup, and Adam

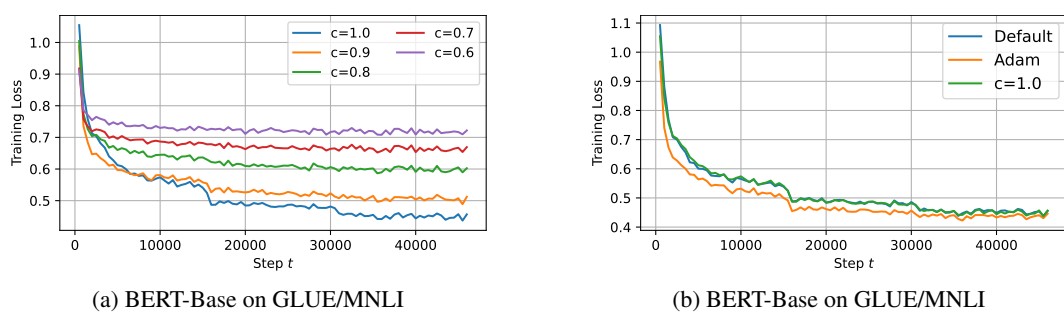

(a) BERT-Base on GLUE/MNLI

(b) BERT-Base on GLUE/MNLI

Figure 3: Training loss vs steps of Adafactor (no update clipping) with different $c$

Table 1: The test accuracy after 3 epochs. We use Adafactor (no update clipping) and Adam to train BERT-Base and BERT-Large on GLUE/MNLI .

|  | $c = 0.6$ | $c = 0.7$ | $c = 0.8$ | $c = 0.9$ | $c = 1.0$ | Default | Adam |
|---|---|---|---|---|---|---|---|
| BERT-Large | 74.78% | 77.32% | 78.90% | 80.65% | 82.28% | 82.35% | 83.28% |
| BERT-Base | 70.08% | 72.91% | 75.56% | 79.68% | 80.24% | 80.64% | 82.56% |

## D.2 Experiments on Adafactor with update clipping

We further test our newly proposed increasing clipping threshold in Theorem 7.1 and compare it with the standard setting where $d_k = 1$. We fix $c = 1$ which is the optimal selection in our theory and use $d_k = k^{\frac{c}{2(\alpha-1)}}$ with $\alpha \in \{2.0, 4.0, 5.0, 7.0, 9.0, 12.0\}$. The other settings keep the same as the ones in Section D.1. We report the training loss curves in Figure 4 and test accuracy in Table 2.

Table 2: The test accuracy after 3 epochs. We use Adafactor with different clipping thresholds to train BERT-Base/Large on GLUE/MNLI.

|  | $\alpha = 2.0$ | $\alpha = 4.0$ | $\alpha = 5.0$ | $\alpha = 7.0$ | $\alpha = 9.0$ | $\alpha = 12.0$ | $d = 1$ |
|---|---|---|---|---|---|---|---|
| BERT-Large | 82.84% | 82.88% | 82.79% | 82.21% | 82.78% | 82.43% | 81.94% |
| BERT-Base | 81.65% | 81.61% | 81.18% | 81.08% | 82.01% | 81.71% | 81.28% |

The results indicate that the increasing clipping thresholds lead to a comparable performance to the constant one as well as Adam. In addition, compared Table 2 with the test accuracy of $c = 1$ in Table 1, it's clear to see that adding update clipping can enhance the performance, particularly when there is no learning rate warm up. This finding is also aligned with the experimental results in [38].

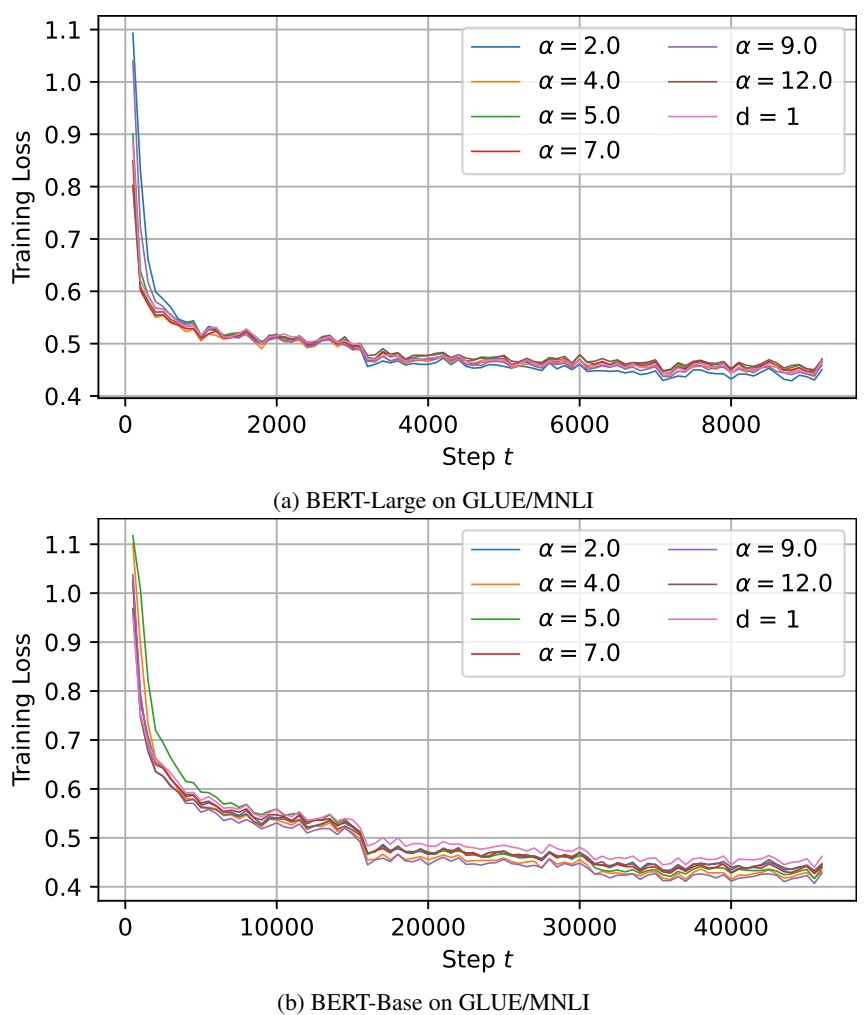

(a) BERT-Large on GLUE/MNLI

(b) BERT-Base on GLUE/MNLI

Figure 4: Training loss vs steps of Adafactor with different update clipping thresholds

