# OpenReview forum: "Theoretical Investigation of Adafactor for Non-Convex Smooth Optimization"
_NeurIPS.cc/2025/Conference — NeurIPS 2025 poster_

### Official Review · Reviewer_zkWc · 2025-06-26

**Clarity:** 2
**Significance:** 2
**Originality:** 2
**Rating:** 4
**Confidence:** 3

**Summary:**

This work establishes convergence guarantees for the Adafactor algorithm in non-convex smooth optimization. Adafactor is analyzed in the deterministic case, the stochastic case without clipping, and in the stochastic case with clipping under a slight modification from a fixed clipping threshold to a time-dependent tuned threshold. Experiments on language tasks are provided to demonstrate the relationship between a theory-related parameter ($c$) and the training loss.

**Questions:**

1. While this work is mainly focused on the dependence on $T$, can the authors compare the established result to other adaptive methods (Such as AdaGrad, Adam, and Shampoo. Not necessarily all of them) in terms of the dependence on the dimension? From what I gather, the dependence is linear in the deterministic case and super-linear in the stochastic case.
2. Can the authors conjecture of other methods whose theoretical analysis might benefit from the tools developed in this work?

Overall, this work contributes to our understanding of adaptive methods by providing convergence guarantees for the Adafactor method, while the established guarantees are yet to indicate why such methods perform so well in practice.

**Ethical Concerns:**

["NO or VERY MINOR ethics concerns only"]

**Final Justification:**

As I mentioned in my review, AdaFactor is somewhat an outdated algorithm, limiting the significance of the result. That said, I still think it may be beneficial for future work, leading me to my limited yet positive support of this work.

**Quality:**

2

**Strengths And Weaknesses:**

**Strengths**

1. It is of value to have proper convergence guarantees for practical methods, in order to better understand the proposed methods and to support future improvements.
2. An analysis of a method such as Adafactor requires handling several technical challenges, which the authors detailed in their work. It is often the case that techniques introduced in one method can help in the analysis of other closely related methods.

**Weaknesses**

3. Beyond establishing the convergence of Adafactor (which is of value of its own), I do not see how the analysis leads to new and meaningful conclusions. This is in addition to the fact that Adafactor, or variants of it, is not a widely used method.
4. A weakness of this work, which often appears in the analysis of such methods, is the sub-optimal bounds (in terms of problem parameters) compared to vanilla SGD. It would be beneficial to demonstrate a somewhat general setting (under additional but reasonable assumptions) where Adafactor has improved convergence compared to SGD. This is minor as this weakness is shared across several works on the topic of adaptive methods, but solving it can contribute to the significance of this work.

---

> ### Author Rebuttal · Authors · 2025-07-29
>
> We appreciate the reviewer’s comments. We address the reviewer’s concerns point by point below:
>
> **Weaknesses.**
>
> > *Beyond establishing the convergence of Adafactor (which is of value of its own), I do not see how the analysis leads to new and meaningful conclusions.*
>
> **Re.** We really appreciate the reviewer’s recognition of the value of our work. Beyond providing the first convergence guarantee for Adafactor, we believe our convergence analysis may yield some interesting conclusions:
>
> - **Comparable convergence rate to Adam:** Although Adafactor is derived as a memory-efficient variant of Adam, our theoretical results could show that its convergence bound matches that of Adam up to logarithmic factors (in terms of $T$) in non-convex smooth optimization setting. Our simple NLP experiments also complement the comparable training performance; see Figure 2(a), Figure 3(b), and Table 1 in Appendix D.1.
> - **Potentially transferable proof techniques:** As Reviewer NfZt points out, *“ideas used here for dealing with the rank-one approximation and clipping of Adafactor can be potentially used for other methods.”*  Our techniques may inspire convergence proofs for other memory-efficient algorithms using matrix approximations, such as GaLore and CAME.
>
> ---
>
> > *This is in addition to the fact that Adafactor, or variants of it, is not a widely used method.*
>
> **Re.** We acknowledge that Adafactor is not as widely used as Adam. Nevertheless, we think that analyzing Adafactor could be an interesting topic for the optimization community. As an early memory-efficient optimizer based on matrix approximation, it has served as a standard baseline in numerous works proposing new memory-efficient algorithms, such as GaLore, CAME, and MicroAdam. Also, several LLMs, including PaLM, T5, UL2, and GShard, have adopted Adafactor as one of their main training optimizers.
>
> ---
>
> > *A weakness of this work, which often appears in the analysis of such methods, is the sub-optimal bounds (in terms of problem parameters) compared to vanilla SGD. It would be beneficial to demonstrate a somewhat general setting (under additional but reasonable assumptions) where Adafactor has improved convergence compared to SGD. This is minor as this weakness is shared across several works on the topic of adaptive methods, but solving it can contribute to the significance of this work.*
>
> **Re.**
> - We acknowledge this point but note that achieving the same (or better) parameter dependencies as SGD may be inherently difficult for Adafactor:
>   - **Theoretical limitation.** It has been shown that SGD is minimax optimal among stochastic first-order methods for non-convex smooth optimization [1]. The convergence improvement is therefore impossible, provided that only smoothness is assumed.
>   - **Trade-off due to memory efficiency.** Adafactor is designed as a memory-efficient variant of Adam. Given the information it discards, it remains unclear whether it's possible to achieve the same or even better parameter dependency as Adam or SGD.
>
> - We agree that it is possible to demonstrate improved convergence performance of Adafactor over SGD under additional but reasonable assumptions. In fact, prior studies have shown that adaptive methods such as AdaGrad and Adam can outperform SGD in certain settings—for example, when the gradient noise is heavy-tailed [2], when the Hessian is highly imbalanced [3, 4], or when the Hessian is nearly block-diagonal [5]. Extending these lines of analysis to Adafactor would be a promising direction for future work.
>
> ---
>
> **Questions.**
>
> > *can the authors compare the established result to other adaptive methods (Such as AdaGrad, Adam, and Shampoo. Not necessarily all of them) in terms of the dependence on the dimension?*
>
> **Re.** Thanks a lot for this valuable suggestion. We will add the following discussion on the dimension dependency:
>
> - After Theorem 5.1: **We can set  $\rho_0 \sim \mathcal{O}((mn)^{-1})$ to derive a convergence bound of $\mathcal{O}(mn)$ with respect to the dimension for full-batch Adafactor.**
> - After Theorem 6.1: **We can set $\rho_0 \sim \mathcal{O}(\mathcal{H}^{-3/2}) \sim \mathcal{O}((mn)^{-3/4})$ given that $H^2 \sim \mathcal{O}(1) $ and $\mathcal{H} \sim \mathcal{O}(\sqrt{mn})$. With the setup, the convergence bound is $\mathcal{O}((mn)^{ 3/4})$ with respect to the dimension for stochastic Adafactor without update clipping.**
> - After Theorem 7.1: **We can set $\rho_0 \sim \mathcal{O}(\mathcal{I}^{-3/2}) \sim \mathcal{O}((mn)^{-3/4})$ given that $I^2 \le \mathcal{O}(1)$ and $\mathcal{I} \sim \mathcal{O}(\sqrt{mn})$. With the setup, the convergence bound is $\mathcal{O}((mn)^{ 3/4})$ with respect to the dimension for stochastic Adafactor with an increasing clipping threshold.**
>
> We will also add the following comparison to some existing results for memory-unconstrained adaptive methods right after Theorem 6.1:
>
> **Under the same conditions of non-convex smoothness, [6] and [7] derive bounds of at least $\mathcal{O}(mn)$ with respect to the dimension for AdaGrad. [8] derives a convergence bound of $\mathcal{O}(mn)$ with respect to the dimension for RMSProp (where they considered a better $\ell_1$-norm). For Adam, many existing works [9, 10, 11] derive $\text{poly}(mn)$ dependency while [12] derive a dimension-free convergence bound. Our convergence bounds show comparable dimension dependency to most results for AdaGrad and Adam, though a gap remains toward achieving fully dimension-free guarantees, and improving the dimension dependency could be further investigated in the future.**
>
> ---
>
> > *Can the authors conjecture of other methods whose theoretical analysis might benefit from the tools developed in this work?*
>
> We suspect that other methods may benefit from our approach in the following points:
>
> - We design a proxy step-size based on the matrix factorization structure of Adafactor to decouple the correlation of adaptive step-size parameters and stochastic gradients. The construction may be beneficial for other memory-efficient optimizers that also apply the matrix factorization, such as GaLore and CAME;
> - We reveal that with a proper setup of clipping threshold, convergence is also guaranteed under the unique update clipping of Adafactor. This update clipping, which uses the root-mean-square to calibrate the second-moment estimator, is different from the standard clipping mechanism. Our techniques may be useful when incorporating this update clipping to other adaptive methods.
> - We use an induction argument to control the unbounded gradient. We believe that it could be used for the analysis of other adaptive methods.
>
> ---
>
> We would be glad to elaborate further if there are any remaining concerns.
>
> **References.**
>
> [1] Lower bounds for Non-Convex Stochastic Optimization, Arjevani, et al., Mathematical Programming, 2023.\
> [2] Why are Adaptive Methods Good for Attention Models? Zhang et al., NeurIPS 2020.\
> [3] Heavy-Tailed Class Imbalance and Why Adam Outperforms Gradient Descent on Language Models, Kunstner et al., NeurIPS 2024.\
> [4] AdaGrad under Anisotropic Smoothness, Liu et al., ICLR 2025.\
> [5] Why Transformers Need Adam: A Hessian Perspective, Zhang et al., NeurIPS 2024.\
> [6] High Probability Convergence of Stochastic Gradient Methods, Liu et al., ICML 2023.\
> [7] Revisiting Convergence of AdaGrad with Relaxed Assumptions, Hong and Lin, UAI 2024.\
> [8] On the $O({d\over T^{1/4}})$ Convergence Rate of RMSProp and Its Momentum Extension Measured by $\ell_1$-Norm, Li et al., 2025.\
> [9] A Simple Convergence Proof of Adam and Adagrad,  Défossez et al., TMLR 2022.\
> [10] Adam Can Converge Without Any Modification On Update Rules, Zhang et al., NeurIPS 2022.\
> [11] Closing the Gap Between the Upper Bound and the Lower Bound of Adam’s Iteration Complexity, Wang et al., NeurIPS 2023.\
> [12] Convergence of Adam Under Relaxed Assumptions, Li et al., NeurIPS 2023.\
> [13] Noise is not the Main Factor behind the Gap between SGD and Adam on Transformers, but Sign Descent might be, Kunstner et al., ICLR 2023.\
> [14] Implicit Bias of Adamw: Norm Constrained Optimization, Xie and Li, ICML 2024.\
> [15] The Implicit Bias of Adam on Separable Data, Zhang et al., arXiv: 2406.10650.

---

> > ### Comment · Reviewer_zkWc · 2025-08-01
> >
> > The you for your response.
> >
> > After reading the other reviews and responses, I keep my limited yet positive support of this work.

---

> > > ### Author Response · Authors · 2025-08-04
> > >
> > > Thanks a lot for your reply! We are willing to further discuss if there is any remaining concern.

---

### Official Review · Reviewer_8wZR · 2025-07-01

**Clarity:** 3
**Significance:** 3
**Originality:** 2
**Rating:** 4
**Confidence:** 3

**Summary:**

The paper provides the theoretical analysis of Adafactor for nonconvex objectives under standard smoothness and stochastic gradient assumptions. The proof techniques are extended from those used to analyze the convergence of earlier adaptive stepsize algorithms such as Adam. The convergence rates are comparable to those of Adam.

**Questions:**

- The introduction before formulation (1) is slightly too narrow. Adafactor and other adaptive stepsize algorithms are proposed to optimize neural networks with multiple matrices instead of only one matrix. Could authors revise the introduction?

- In Figure 1, could authors also compare Adam's training curve with Adafactor's? This would convince the reader about the effectiveness of the implementation of Adafactor.

**Ethical Concerns:**

["NO or VERY MINOR ethics concerns only"]

**Final Justification:**

Authors have addressed my concerns regarding $\epsilon$ in the rebuttal. Authors also acknowledge that the problem formulation could be more general. I thus keep my support for the paper.

**Limitations:**

yes

**Paper Formatting Concerns:**

No concern

**Quality:**

3

**Strengths And Weaknesses:**

Strengths

- Adafactor is an important memory-saving adaptive stepsize algorithm, and its theoretical analysis would be important for practitioners and theoreticians as well.

- The writing is clear. Assumptions and theoretical results are clearly stated. The new techniques are properly highlighted in Section 8.

Weakness

- The value of $c_0$, which is related to $\epsilon_1$ according to Theorem 6.1, should be the order of $10^{-30}$ based on [37]. This may result in super large constants $H$ and $I$. Authors may also discuss the theoretical implications of small $\epsilon_1$.

-  One motivation of Adafactor is reducing memory cost from $O(mn)$ to $O(m+n). Therefore, it is better to also make the dependence on $m$ and $n$ explicit in Theorem 5.1, Theorem 6.1, and Theorem 7.1.

---

> ### Author Rebuttal · Authors · 2025-07-29
>
> We appreciate the reviewer’s comments. We address the reviewer’s concerns point by point below:
>
> > *The value of $c_0$ , which is related to $\epsilon_1$ according to Theorem 6.1, should be the order of $10^{-30}$ based on [37]. This may result in super large constants $H$ and $I$ .*
>
> **Re.** We appreciate the reviewer’s careful observation. Note that the relationship between $\epsilon_1$ and $c_0$ is:
>
> - In Theorem 5.1, we require $\epsilon_1 = \frac{c_0}{mn}$;
> - In Theorems 6.1 and 7.1, we require $\epsilon_1 = \frac{c_0}{\sqrt{mn}}$.
>
> Since the dimension $mn$ could be large in some practical examples, $\epsilon_1$ will naturally be small even when $c_0$ is a moderate constant. This allows us to set $c_0$ as a small, fixed constant without needing to choose it on the order of $10^{-30}$. Our theoretical setup for $\epsilon_1$ suggests the use of a small value that could be roughly in line with the typical setting, though there may be a gap between the theoretical and practical setups. We will make this clearer in the revision.
>
> ---
>
> > *Therefore, it's better to also make the dependence on $m$ and $n$ explicit in Theorems 5.1, 6.1 and 7.1.*
>
> **Re.** We appreciate this valuable suggestion. We will add the following discussion on the dimension dependency:
>
> - After Theorem 5.1: **We can set  $\rho_0 \sim \mathcal{O}((mn)^{-1})$ to derive a convergence bound of $\mathcal{O}(mn)$ with respect to the dimension for full-batch Adafactor.**
> - After Theorem 6.1: **We can set $\rho_0 \sim \mathcal{O}(\mathcal{H}^{-3/2}) \sim \mathcal{O}((mn)^{-3/4})$ given that $H^2 \sim \mathcal{O}(1) $ and $\mathcal{H} \sim \mathcal{O}(\sqrt{mn})$. With the setup, the convergence bound is $\mathcal{O}((mn)^{ 3/4})$ with respect to the dimension for stochastic Adafactor without update clipping.**
> - After Theorem 7.1: **We can set $\rho_0 \sim \mathcal{O}(\mathcal{I}^{-3/2}) \sim \mathcal{O}((mn)^{-3/4})$ given that $I^2 \sim \mathcal{O}(1)$ and $\mathcal{I} \sim \mathcal{O}(\sqrt{mn})$. With the setup, the convergence bound is $\mathcal{O}((mn)^{ 3/4})$ with respect to the dimension for stochastic Adafactor with an increasing clipping threshold.**
>
> We will also add the following comparison to some existing results for memory-unconstrained adaptive methods right after Theorem 6.1:
>
> **Under the non-convex smoothness optimization setting, [1] and [2] derive convergence bounds of at least $\mathcal{O}(mn)$ with respect to the dimension for AdaGrad. [3] derives a convergence bound of $\mathcal{O}(mn)$ with respect to the dimension for RMSProp (where they considered $\ell_1$-norm as the measurement). For Adam, many existing works [4, 5, 6] derive convergence bounds with a $\mathcal{O}(\text{poly}(mn))$ dependency, while [7] derives a dimension-free convergence bound. Our convergence bounds show comparable dimension dependency to most results for AdaGrad and Adam, though a gap remains toward achieving fully dimension-free guarantees, and improving the dimension dependency could be further investigated in the future.**
>
> ---
>
> > *The introduction before formulation (1) is slightly too narrow. Adafactor and other adaptive stepsize algorithms are proposed to optimize neural networks with multiple matrices instead of only one matrix. Could the authors revise the introduction?*
>
> **Re.** We really appreciate the reviewer for pointing this out. In Formula (1), we follow the standard setup in stochastic optimization where $X$ represents **all** the trainable weights in a model, e.g., the combination of weight matrices in each layer of a deep neural model. We will add the following statement after Formula (1) to clarify this:
>
> **where $X$ denotes all the trainable weights of the model**.
>
> We also recognize that Adafactor applies its matrix factorization separately to each weight matrix (e.g., per-layer weight matrices) when training multi-layer models. Therefore, a more precise interpretation is that $X$ consists of a set of matrices $\\{X^{(i)}\\}_{i=1}^L$, where each $X^{(i)} \in \mathbb{R}^{m_i \times n_i}$ corresponds to a layer-wise weight matrix.
>
> Our theoretical analysis treats $X$ as a unified matrix in $\mathbb{R}^{m \times n}$ to simplify the convergence analysis while still capturing the essential behavior of Adafactor. We believe that it would be an interesting direction for future work to analyze convergence under a per-matrix (i.e., layer-wise) factorization setting, which more closely reflects the practical implementation in training multi-layer models.
>
> ---
>
> > *In Figure 1, could authors also compare Adam's training curve with Adafactor's? This would convince the reader about the effectiveness of the implementation of Adafactor.*
>
> **Re.** Thanks a lot for this helpful suggestion. We have included the comparison of Adafactor (with decay rate $c=1.0$) with Adam in Figure 2(a), Figure 3(b) and Table 1 from Appendix D.1. We will add Adam's training loss curve in Figure 1 in the revised version to make the comparison more direct and accessible.
>
> ---
>
> We would be glad to elaborate further if there are any remaining concerns.
>
> **References.**
>
> [1] High Probability Convergence of Stochastic Gradient Methods, Liu et al., ICML 2023.\
> [2] Revisiting Convergence of AdaGrad with Relaxed Assumptions, Hong and Lin, UAI 2024.\
> [3] On the $O({d\over T^{1/4}})$ Convergence Rate of RMSProp and Its Momentum Extension Measured by $\ell_1$-Norm, Li et al., 2024.\
> [4] A Simple Convergence Proof of Adam and Adagrad,  Défossez et al., TMLR 2022.\
> [5] Adam Can Converge Without Any Modification On Update Rules, Zhang et al., NeurIPS 2022.\
> [6] Closing the Gap Between the Upper Bound and the Lower Bound of Adam’s Iteration Complexity, Wang et al., NeurIPS 2023.\
> [7] Convergence of Adam Under Relaxed Assumptions, Li et al., NeurIPS 2023.

---

> > ### Comment · Reviewer_8wZR · 2025-08-05
> > **Thank authors for the response**
> >
> > My concerns are addressed or will be addressed in the revision. I will maintain my score to support the manuscript.

---

> > > ### Author Response · Authors · 2025-08-05
> > > **Thank Reviewer 8wZR**
> > >
> > > Thank you once again for your time and valuable feedback, which has improved our paper.

---

> ### Comment · Area_Chair_De28 · 2025-08-05
>
> Dear Reviewer,
>
> To facilitate further evaluation of the submission, please take a moment to respond to the authors’ rebuttal.
>
> Best,
>
> AC

---

### Official Review · Reviewer_NfZt · 2025-07-02

**Clarity:** 3
**Significance:** 3
**Originality:** 3
**Rating:** 5
**Confidence:** 3

**Summary:**

The paper studies the convergence of the Adafactor algorithm for smooth non-convex functions. It provides convergence rates under sub-Gaussian noise that match the optimal rate up to log factors for this class of functions.

**Questions:**

-  Can your approach work with smoothness with operator norms instead of Frobenius?

- What about other smoothness assumptions like $(L_0,L_1)$?

**Ethical Concerns:**

["NO or VERY MINOR ethics concerns only"]

**Limitations:**

Maybe more general smoothness assumptions and operator norms.

**Quality:**

3

**Strengths And Weaknesses:**

Strengths:
- The paper is clearly written and does a good job introducing the algorithm for readers who might not be super familiar, and it also does a good job explaining the challenges that were faced in obtaining such theoretical results.
- The theoretical analysis is interesting for the community: ideas used here for dealing with the one rank approximation and the clipping of Adafactor can be potentially used for other methods.

Weaknesses (minor):
- The theory assumes smoothness and bounds the noise in terms of the Frobenius norm, this norm hides (in the worst case) a factor that is proportional to the dimension compared to operator norms (associated to $\ell_2$ for example).
- I can also go on about smoothness that is known not to hold for NNs, but we have to start somewhere. It would be nice if this could be extended to more general smoothness assumptions.

---

> ### Author Rebuttal · Authors · 2025-07-29
>
> We appreciate the reviewer’s comments. We address the reviewer’s concerns point by point below:
>
> > (a). *The theory assumes smoothness and bounds the noise in terms of the Frobenius norm; this norm hides (in the worst case) a factor that is proportional to the dimension compared to operator norms (associated to $\ell_2$-norm for example).* (b). *Can your approach work with smoothness with operator norms instead of Frobenius?*
>
> **Re.** We thank the reviewer for this insightful question. Note that the smoothness assumption with respect to a general norm is:
>
> $$
> \\|\nabla f(Y) - \nabla f(X)\\| \le L\\|Y-X\\|_{*},
> $$
>
> where $\\|\cdot \\|$ and $\\|\cdot \\|_{*}$ are dual norms. In this paper, we consider the smoothness with standard Frobenius norm $\\|\cdot\\|_F$, which is self-dual, and it is aligned with the standard smoothness definition in terms of $\ell_2$-norm in the stochastic optimization.
>
> We agree that using the operator norm could be meaningful in some aspects. For instance, by adopting the induced $\ell_2$-norm $\\|\cdot \\|_{2 \rightarrow 2}$, the dual norm is the nuclear norm.  In this case, one could establish a descent lemma based on the operator norm and potentially follow similar proof techniques in our paper to derive a convergence guarantee. However, for this moment, the answer is not very clear, which deserves further study.
>
> We also kindly remind that, although the operator norm could avoid the dimension, the dimension factor may emerge in its dual norm. For example, the nuclear norm may also introduce implicit dependencies on the dimensions without making additional assumptions.
>
> ---
>
> > (a). *I can also go on about smoothness that is known not to hold for NNs, but we have to start somewhere. It would be nice if this could be extended to more general smoothness assumptions.* (b). *What about other smoothness assumptions like $(L_0,L_1)$?*
>
> **Re.** We agree with that the standard $L$-smoothness assumption may not always hold for neural networks in some practical cases. Developing theory to more general smoothness conditions—such as the $(L_0,L_1)$-smoothness [1] is indeed valuable. However, such generalizations could be nontrivial, especially because the gradient norm can be unbounded and thus may complicate convergence analysis. We think that the point can be a promising direction for future work, which will be added in **Limitation** part as follows:
>
> **The convergence results for Adafactor are established under the standard smoothness assumption. It would be interesting to further investigate convergence under more general smoothness conditions that better reflect practical applications, such as $(L_0, L_1)$-smoothness.**
>
> ---
>
> We would be glad to elaborate further if there are any remaining concerns.
>
> **References.**
>
> [1] Why Gradient Clipping Accelerates Training: A Theoretical Justification for Adaptivity, Zhang et al., ICLR 2020.\
> [2] The Geometry of Sign Gradient Descent, Balles et al., arXiv 2020.\
> [3] Understanding the Generalization of Adam in Learning Neural Networks with Proper Regularization, Zou et al.,arXiv 2021.

---

> ### Comment · Area_Chair_De28 · 2025-08-05
>
> Dear Reviewer,
>
> To facilitate further evaluation of the submission, please take a moment to respond to the authors’ rebuttal.
>
> Best,
>
> AC

---

### Official Review · Reviewer_qPLs · 2025-07-14

**Clarity:** 2
**Significance:** 2
**Originality:** 2
**Rating:** 4
**Confidence:** 3

**Summary:**

The paper analyzes the convergence rate to a stationary point of Adafactor.

**Questions:**

# Suggestions
1. Some symbols look too close to each other, I suggest changing some of them.
    1. $c$ and $c_0$ have different purposes, but almost identical symbols.
    2. $G_k$ and $\mathbf{G}_k$ looks almost identical. I think that bold text is not enough to differentiate between symbols.
2. The corollaries are not a direct result of the theorem as they use different $\beta_{2,k}$. I think the paper should clarify that the proofs of the corollaries are in the appendix.

**Ethical Concerns:**

["NO or VERY MINOR ethics concerns only"]

**Final Justification:**

The authors have addressed all the points I mentioned.
However, the paper only analyzes the convergence rate to a stationary point of Adafactor. This was already analyzed for other algorithms, thus limiting the theoretical significance of this paper.

**Limitations:**

yes

**Quality:**

3

**Strengths And Weaknesses:**

# Strengths
1. For the deterministic case, the paper proves an optimal convergence rate $\mathcal{O}(1/T)$ for Adafactor.
2. For the stochastic case, the paper proves a near-optimal convergence rate $\mathcal{O}(1/\sqrt{T})$ for Adafactor.

# Weaknesses
1. Some things are not written in the most clear way. More details are written in the suggestions.

---

> ### Author Rebuttal · Authors · 2025-07-29
>
> We appreciate the reviewer’s comments. We address the reviewer’s concerns point by point below:
>
> > *Some symbols look too close to each other, I suggest changing some of them.*
>
> **Re.** We will revise some symbols that may be confused by readers. Particularly, for what the reviewer suggests on $c,c_0$ and $G_k,{\bf G}_k$:
>
> - We will change the decay rate parameter notation from $c$ to $\gamma$, e.g., in Formula (7), Line 179, and Line 206;
> - We will use ${\bf G}_k$ to denote the stochastic gradient at $k$-th iteration and change the notation $G_k$ to $D_k:=\max\_{j \in [k]}\\|\nabla F(X\_k)\\|\_F$, e.g., in Line 224 and Line 467.
>
> ---
>
> > *I think the paper should clarify that the proofs of the corollaries are in the appendix.*
>
> **Re.** We will add the following statement in the revised version:
>
> - ‘**The detailed proof can be found in Appendix B.2.**’ in Line 194;
> - ‘**The detailed proof can be found in Appendix C.4.**’ in Line 223.
>
> ---
>
> We would be glad to elaborate further if there are any remaining concerns.

---

> > ### Comment · Reviewer_qPLs · 2025-08-05
> >
> > I want to thank the authors for their response.
> > For now, I will keep my score as it is.

---

> > > ### Author Response · Authors · 2025-08-05
> > > **Thank Reviewer qPLs**
> > >
> > > Thank you once again for your time and valuable feedback, which has improved our paper.

---

> ### Comment · Area_Chair_De28 · 2025-08-05
>
> Dear Reviewer,
>
> To facilitate further evaluation of the submission, please take a moment to respond to the authors’ rebuttal.
>
> Best,
>
> AC

---

### Note · Authors · 2025-08-15

We would like to sincerely thank the Area Chair and all reviewers for their thoughtful feedback and efforts in handling our manuscript.

We are grateful for the reviewers' positive scores and kind support.

Our work provides, to the best of our knowledge, the first convergence guarantees for Adafactor in non-convex smooth optimization, including optimal rates in the deterministic case and near-optimal rates under sub-Gaussian noise. We believe that the analysis is challenging, largely due to rank-one approximation and time-dependent clipping, and our techniques may be applicable to other adaptive methods.

We have carefully considered and addressed the raised concerns (that are mostly on the presentation clarity and exploring further discussions) during the Rebuttal and Discussion periods:

- **Reviewer qPLs** noted notation and presentation issues. In response, we have revised some similar symbols accordingly and clarified that corollaries have separate proofs in the appendix.
- **Reviewer NfZt** raised a minor concern about the potential hidden dimension factor in the Frobenius-norm smoothness assumption. In response, we claimed that this setting could align with standard vectorized formulations in stochastic optimization and facilitate comparisons with existing Adam/AdaGrad results. Also, we have discussed possible extensions to $(L_0,L_1)$-smoothness accordingly.
- **Reviewer 8wZR** raised concerns about the setup of $\epsilon_1$ and suggested discussing the dimension dependency. In response, we have explained that $\epsilon_1$ scales inversely with the dimension, which helps keep $H$ and $I$ moderate compared to the total iteration number $T$. We also made the dimensional dependence explicit in Theorems 5.1–7.1 accordingly.
- **Reviewer zkWc** raised concerns about the broader impact of the work, as well as the suboptimal convergence bound (in terms of problem parameters) compared to SGD. While Adafactor is indeed not as widely used as Adam, it remains a commonly used baseline for memory-efficient optimizers. We hope that our results can serve as a starting point for further studies, including settings where Adafactor may outperform SGD under further assumptions.

We sincerely appreciate the reviewers’ valuable suggestions, which have helped us improve the presentation and clarity of our work.

---

### Decision · Program_Chairs · 2025-09-17

**Decision:**

Accept (poster)

**Comment:**

This paper presents the first convergence analysis of Adafactor in the setting of nonconvex smooth optimization. To the best of my knowledge, such a result is novel and a timely contribution. The analysis is clearly written and technically sound, with the authors providing satisfactory responses to several technical concerns raised during the review process. There are two key limitations: (1) the theoretical results do not explicitly demonstrate the advantage of Adafactor in terms of convergence rate compared to existing optimizers; (2) the assumption of smoothness may not hold for neural networks. Nonetheless, given the originality of the contribution, the clarity of exposition, and the addressed reviewer feedback, I recommend acceptance. The authors are expected to discuss the limitations explicitly in the final version of this paper.